# Repeated introductions and intensive community transmission fueled a mumps virus outbreak in Washington State

Louise H Moncla[1†]*, Allison Black[1,2†], Chas DeBolt[3], Misty Lang[3], Nicholas R Graff[3], Ailyn C Pérez-Osorio[3], Nicola F Müller[1], Dirk Haselow[4], Scott Lindquist[3], Trevor Bedford[1,2]*

[1]Vaccine and Infectious Disease Division, Fred Hutchinson Cancer Research Center, Seattle, United States; [2]Department of Epidemiology, University of Washington, Seattle, United States; [3]Office of Communicable Disease Epidemiology, Washington State Department of Health, Shoreline, United States; [4]Arkansas Department of Health, Little Rock, United States

**Abstract** In 2016/2017, Washington State experienced a mumps outbreak despite high childhood vaccination rates, with cases more frequently detected among school-aged children and members of the Marshallese community. We sequenced 166 mumps virus genomes collected in Washington and other US states, and traced mumps introductions and transmission within Washington. We uncover that mumps was introduced into Washington approximately 13 times, primarily from Arkansas, sparking multiple co-circulating transmission chains. Although age and vaccination status may have impacted transmission, our data set could not quantify their precise effects. Instead, the outbreak in Washington was overwhelmingly sustained by transmission within the Marshallese community. Our findings underscore the utility of genomic data to clarify epidemiologic factors driving transmission and pinpoint contact networks as critical for mumps transmission. These results imply that contact structures and historic disparities may leave populations at increased risk for respiratory virus disease even when a vaccine is effective and widely used.

*For correspondence:
lhmoncla@gmail.com (LHM);
tbedford@fredhutch.org (TB)

†These authors contributed equally to this work

Competing interests: The authors declare that no competing interests exist.

## Introduction

In 2016 and 2017, mumps virus swept the United States in the country's largest outbreak since the pre-vaccine era (*CDCMMWR, 2019*). Washington State was heavily affected, reporting 889 confirmed and probable cases. Longitudinal studies (*Davidkin et al., 2008*), epidemiologic outbreak investigations (*Cardemil et al., 2017*), and epidemic models (*Lewnard and Grad, 2018*) suggest that mumps vaccine-induced immunity wanes over 13–30 years, consistent with the preponderance of young adult cases in recent outbreaks. Like with other recent mumps outbreaks, most Washington cases in 2016/17 were vaccinated. Unusually though, while most US outbreaks in 2016/2017 were associated with university settings (*Albertson et al., 2016*; *Bonwitt et al., 2017*; *Donahue et al., 2017*; *Golwalkar et al., 2018*; *Iowa Mumps Outbreak Response Team et al., 2018*; *Wohl et al., 2020*), incidence in Washington was highest among children aged 10–18 years, younger than expected given waning immunity. The outbreak was also peculiar in that approximately 52% of the total cases were Marshallese, an ethnic community that comprises ~0.3% of Washington's population. These same phenomena were also observed in Arkansas. Of the 2954 confirmed and probable Arkansas cases, 57% were Marshallese, and 57% of cases were children aged 5–17 (*Fields et al., 2019*). Among the infected school-aged children in Arkansas and Washington, >90% had previously received two doses of MMR vaccine (*Fields et al., 2019*). The high proportion of vaccinated cases,

younger-than-expected age at infection, disproportionate impact on the Marshallese community, and epidemiologic link to Arkansas suggest that factors beyond waning immunity are necessary to explain mumps transmission during this outbreak in Washington.

The US and the Marshall Islands are closely linked through a history that continues to impact the health of US-residing Marshallese to this day. Between 1947 and 1986, the United States occupied the Republic of Marshall Islands and detonated the equivalent of >7000 Hiroshima size nuclear bombs as part of its nuclear testing program (*Barker, 2012*). The effects were devastating, precipitating widespread environmental destruction, nuclear contamination, and dire health consequences (*Hallgren et al., 2015*; *Niedenthal, 1997*; *Palafox et al., 2007*; *Simon, 1997*; *Takahashi et al., 1997*). Marshallese individuals inhabiting the targeted atolls were forcibly moved to other islands, and many were exposed to nuclear fallout (*Abella et al., 2019*) that persists on the Islands today (*Bordner et al., 2016*). Significant concern remains within the community regarding long-term health impacts of nuclear exposure and its potential impacts on immune function. Marshallese individuals living on and off the Islands experience significant health disparities including a higher burden from infectious diseases and chronic health conditions (*Adams et al., 1986*; *Wong et al., 1979*; *Yamada et al., 2004*). Compounding these disparities, from 1996 to 2020 (*Hirono, 2019*), Marshallese individuals were specifically excluded from Medicaid eligibility despite legal residency in the US permitted under the Compact of Free Association (COFA) Treaty. As a result, many US-residing Marshallese are uninsured, with poor access to healthcare (*McElfish et al., 2015*). Marshallese households are more likely to be multigenerational and tend to be larger on average (*Harris and Jones, 2005*; *US Census Bureau, 2021*), potentially increasing the number and intensity of interactions among individuals. These factors combined mean that Marshallese individuals may be at increased risk of respiratory virus infection.

Clarifying the determinants of infectious disease transmission is important for prioritizing prevention and mitigation resources. However, sampling bias presents a persistent challenge for elucidating source-sink dynamics from genomic data (*De Maio et al., 2015*; *Dudas et al., 2018*; *Frost et al., 2015*; *Kühnert et al., 2011*; *Lemey et al., 2020*; *Stack et al., 2010*), which may undermine the utility of genomic epidemiological studies in some situations. Here, we formulate a set of genomic epidemiological approaches that are robust to sampling frame and apply them to investigate patterns of mumps transmission in Washington. We sequenced 110 mumps viral genomes obtained from specimens collected from laboratory-confirmed mumps cases in Washington State and another 56 from other US states collected between 2006 and 2018. We employ a novel application of phylogeographic methods to detailed epidemiologic data on age, vaccination status, and community membership, and develop a new statistic for quantifying transmission in the tree. By combining these phylodynamic approaches with community health advocate interviews that contextualize our results, we provide a framework for investigating viral transmission dynamics that is sensitive to community health priorities and readily applicable to other viral pathogens.

## Results

### Outbreak characteristics and data set composition

We generated genome sequences for 110 PCR-positive mumps samples collected throughout Washington State during 2016/2017, and 56 samples collected in Wisconsin, Ohio, Missouri, Alabama, and North Carolina between 2006 and 2018 (*Supplementary file 1a*). The Washington State outbreak began in October 2016 and peaked in winter of 2017, culminating in 889 confirmed and probable cases across Washington (*Figure 1*). Individuals aged <1 to 64 years were affected, but incidence was highest among children aged 10–14 (44.9 cases per 100,000) and 15–19 (47.0 per 100,000) (*Supplementary file 1b*). Among outbreak cases 5–19 years of age, 91% of individuals were considered up-to-date on mumps vaccine. Adults in the age group most likely to be parents of school aged children (20–39 years old) were infected at a rate of only 12.9 cases per 100,000, but comprised a significant proportion (29%) of total cases (*Supplementary file 1b*). While Marshallese individuals comprise only ~0.3% of Washington's total population, they accounted for 52% of reported mumps cases (*Supplementary file 1c*). Among Marshallese cases aged 5–19, 93% were up-to-date on vaccination, suggesting that this over-representation is not attributable to poor vaccine coverage.

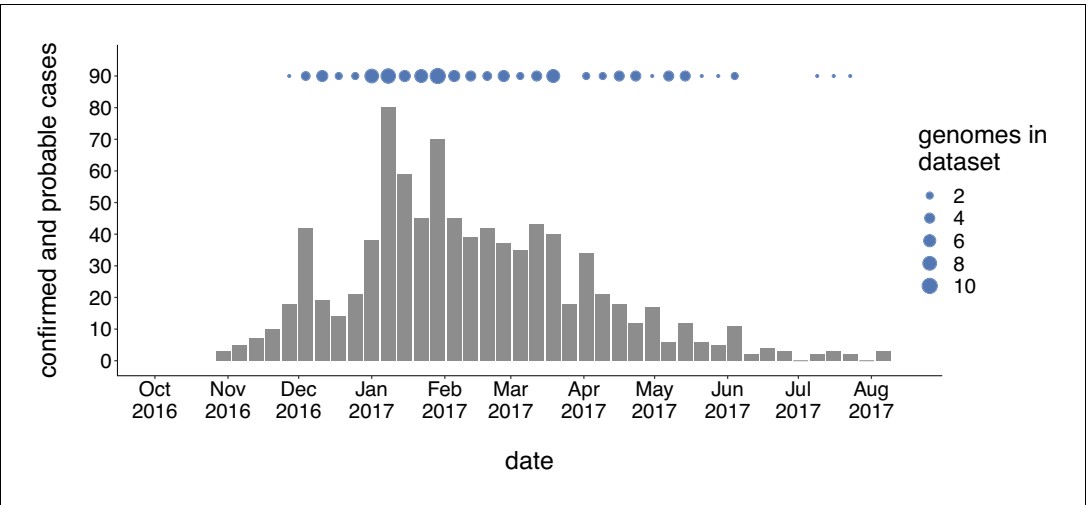

**Figure 1.** Genomic sampling covers the duration of the outbreak. The first mumps case in Washington was reported on October 30, 2016, and case counts peaked in the winter of 2017. Here we show recorded numbers of confirmed and probable cases by epidemiologic (epi) week. Blue dots above the epidemiologic curve represent the number of Washington genome sequences sampled from viruses collected during that epi week.

The online version of this article includes the following source data for figure 1:

**Source data 1.** Washington State mumps case counts in 2016–2017.

**Source data 2.** Metadata for sequences generated in this manuscript with collection dates.

## Outbreaks across North America are related

We combined our sequence data with publicly available full genome sequences sampled from North America between 2006 and 2018 and built a time-resolved phylogeny, inferring migration history among 27 US states and Canadian provinces (*Figure 2*, *Figure 2—figure supplement 1*; *Figure 2— figure supplement 2*). Sequences from samples collected between 2006 and 2014 clustered with other North American mumps viruses sampled from the same times. Nine Washington sequences were highly divergent from other North American genotype G viruses, with a time to the most recent common ancestor (TMRCA) of ~22 years (*Figure 2*, blue tips with long branches clustered toward the top of the tree). To place these genomes in context, we built a divergence tree using all publicly available global full genome mumps sequences (*Figure 2—figure supplement 3*). Seven of these divergent Washington sequences cluster with viruses sampled from New Zealand (*Figure 2— figure supplement 3*), suggesting they could be travel-related. The other two sequences cluster with other divergent genotype G viruses sampled from geographically disparate locations (*Figure 2—figure supplement 3*). The remaining Washington sequences nest within the diversity of other North American viruses, and descend from the same mumps lineage that has circulated in North America since 2006 (*Figure 2*). We observe substantial geographic mixing along the tree. While viruses from the northeast (teal tips and branches) seeded outbreaks in the Northeast and Midwest, we also infer transmission from the Northeast to Southern US states and British Columbia. Despite the close geographic proximity between British Columbia and Washington, most British Columbia sequences form a distinct cluster on a long branch (*Figure 2*), suggesting seeding from an unsampled location. Although viruses from Washington are scattered throughout the phylogeny, most cluster within a clade of viruses sampled in Arkansas (*Figure 2*).

Mumps is classified into 12 genotypes (labeled A–N, excluding E and M) based on its SH gene sequence. There is some evidence that mumps genotypes are geographically associated (*Nomenclature, 2020*), and the vast majority of mumps viruses circulating in North America since 2006 have been genotype G viruses. Although most samples in our data set are also genotype G, we did sequence three viruses that group in different genogroups. One sample from Wisconsin in 2006 grouped with genotype A viruses, another sample from Wisconsin in 2015 grouped with genotype H viruses, and one sample from Washington in 2017 grouped with genotype K viruses (*Figure 2—*

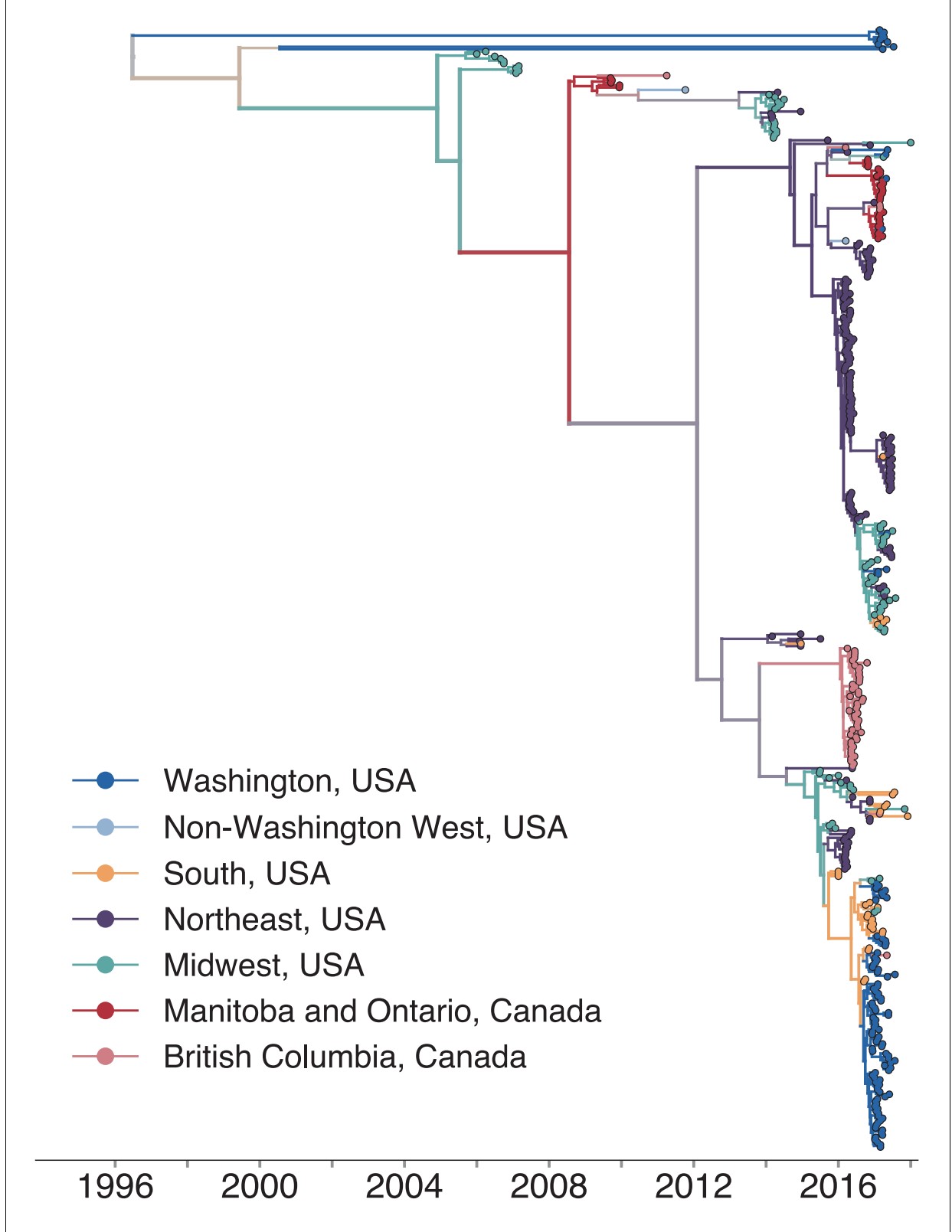

**Figure 2.** North American mumps outbreaks are related. We combined all publicly available North American mumps genomes and built a time-resolved phylogeny. We inferred geographic transmission history between each US state and Canadian province using a discrete trait model, but have grouped these locations into regions for plotting purposes. A tree colored by the full geographic transmission history across all 27 locations is shown in *Figure 2—figure supplement 2*. Here, we display the maximum clade credibility tree, where color represents geographic location. We grouped the US

*Figure 2 continued on next page*

*Figure 2 continued*

states by geography as follows: non-Washington West include California and Montana; Midwest USA includes North Dakota, Kansas, Missouri, Iowa, Wisconsin, Indiana, Michigan, Ohio, and Illinois; South USA includes North Carolina, Alabama, Virginia, Georgia, Texas, Arkansas, and Louisiana; Northeast USA includes New York, Massachusetts, Pennsylvania, New Hampshire, and New Jersey. Canadian provinces are also grouped by geographic area. The x-axis represents the collection date (for tips), or the inferred time to the most recent common ancestor (for internal nodes). The internal node coloring represents the sum of the posterior probabilities for each inferred geographic division within the most probable region. For example, since we group Manitoba and Ontario into the same Canadian region, if a node was inferred with highest probability to circulate in Manitoba, then the node would be colored red to represent that Canadian region. The opacity of the color then corresponds to the sum of the probabilities that the node circulated in Manitoba or that the node circulated in Ontario. The posterior probability is expressed by the color gradient, where increasingly gray tone represents decreasing certainty of the inferred geographic state. The ancestral state at the root was poorly resolved and is therefore colored mostly gray.

The online version of this article includes the following source data and figure supplement(s) for figure 2:

**Source data 1.** XML file to run discrete trait phylogeographic analysis of North American mumps transmission shown in *Figure 2*, with combined mcc tree and output log files.

**Source data 2.** Divergence trees with metadata for divergence trees shown in *Figure 2—figure supplement 4* and *Figure 2—figure supplement 5*.

**Figure supplement 1.** Mumps genomes accumulate mutations linearly over time.

**Figure supplement 2.** Phylogeographic history inferred with the full, 27-state discrete trait model.

**Figure supplement 3.** Placement of divergent Washington and non-genotype G genomes on a global phylogeny.

**Figure supplement 4.** The full genome divergence tree closely matches the time-resolve phylogeny.

**Figure supplement 5.** SH gene sequences are inadequate for fine-scale resolution of mumps transmission.

---

*figure supplement 3*). The Washington genotype K virus (Washington.USA/9.17/FH94/K) is closely related to a genotype K mumps virus collected during a mumps outbreak in Massachusetts from an individual who reported international travel (*Wohl et al., 2020*; *Figure 2—figure supplement 3*). These divergent, non-genotype G genomes were excluded from further phylogenetic analysis.

## Mumps was introduced into Washington multiple independent times

Estimating the number and timing of viral introductions is important for estimating epidemiologic parameters and evaluating public health surveillance systems, but detecting these dynamics may be challenging with case count data alone (*Faria et al., 2017*; *Grubaugh et al., 2017*). The Washington Department of Health had identified a single potential index case infected in October 2016. To determine whether the genomic data similarly supported a single introduction of mumps to Washington state, we separated each introduction inferred in the maximum clade credibility tree and plotted each as its own transmission chain (*Figure 3a*). We enumerated the number of transitions into Washington in each tree in the posterior set, and plotted the distribution of Washington introductions consistent with the phylogeny (*Figure 3b*).

Genomic data show that mumps was introduced into Washington State approximately 13 independent times (95% highest posterior density, HPD: 12–15), from geographically disparate locations (*Figure 3*). In addition to the nine highly divergent Washington tips (*Figure 2—figure supplement 3*), we detect one introduction from Massachusetts that descends from a long branch. Prior to being sampled in Washington, this lineage was last inferred to circulate in Massachusetts in late 2015. Thus rather than representing a direct introduction from Massachusetts to Washington, this lineage likely moved through other geographic locations that lack genomic sampling. We infer introductions from Ontario and Missouri that each lead to one to three sampled cases (*Figure 3b*), suggesting limited onward transmission following these introductions. In contrast, four introductions from Arkansas account for 92/110 sequenced cases, suggesting that these introductions led to more sustained chains of transmission following introduction (*Figure 3b*). We refer to the largest cluster as the 'primary outbreak clade', and infer its introduction from Arkansas to Washington around August 2016 (August 7, 2016, 95% HPD: July 11, 2016 to September 19, 2016, *Figure 3b*), 3.5 months before Washington's first reported case. These data reveal that what had appeared to be a single outbreak based on case surveillance data was in fact a series of multiple introductions, primarily from Arkansas, sparking overlapping and co-circulating transmission chains.

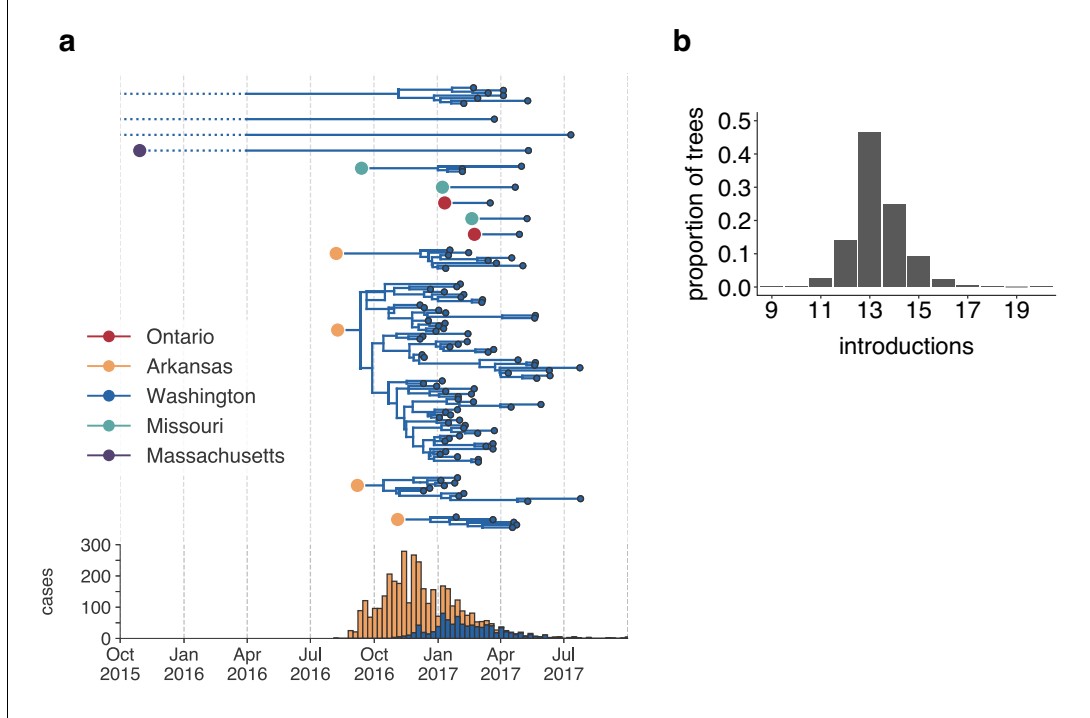

**Figure 3.** The mumps outbreak in Washington was seeded by approximately 13 introductions. (**a**) We separated each introduction into Washington inferred on the maximum clade credibility tree (*Figure 2*) and plotted them independently. Large, colored dots represent the inferred geographic location that the Washington introduction was seeded from. Branches that extend further back in time than April 2016 (approximately 6 months prior to the first reported case in Washington) are dotted to represent that transmission likely occurred via other, unsampled locations. The nine tips at the very top without inferred geographic ancestry represent the highly divergent nine Washington genomes with a TMRCA of ~22 years that are highlighted in *Figure 2—figure supplement 3* and shown in *Figure 2*. For reference, the cumulative case counts from Arkansas and Washington are plotted below. (**b**) For each tree in the posterior set, we inferred the number of introductions into Washington. We plot the proportion of trees in the posterior set in which that number of introductions was inferred.

The online version of this article includes the following source data for figure 3:

**Source data 1.** Inferred introductions into Washington State across posterior distribution.

## SH gene sequencing is insufficient for fine-grained geographic inference

Mumps virus surveillance and genotyping rely on the SH gene (*Centers for Disease Control and Prevention, 2019a*), a short, 316 bp gene that is simple and rapid to sequence. To determine whether SH gene sequencing would have produced similar results, we built a divergence tree using our set of North American full genomes (*Figure 2—figure supplement 4*), and then truncated that data to include only SH gene sequences (*Figure 2—figure supplement 5*). Almost all North American SH genes were identical, resulting in a single, large polytomy (*Figure 2—figure supplement 5*). This indicates that SH sequences lack sufficient resolution to elucidate fine-grained patterns of geographic spread, consistent with previous findings (*Gouma et al., 2016*; *Wohl et al., 2020*).

## Quantifying differences in transmission patterns within Washington

In both Arkansas and Washington, Marshallese individuals comprised over 50% of mumps cases, despite accounting for a much lower proportion of the population in both states. Phylogenetic reconstruction links the outbreaks in Washington and Arkansas, placing most sampled mumps genomes in Washington as descendant from Arkansas. We sought to investigate how mumps transmission may have differed within Marshallese and non-Marshallese communities within the same outbreak. Phylogenetic trees reflect the transmission process and can be used to quantify differences in transmission patterns among population groups. If transmission rates were distinct between Marshallese and non-Marshallese mumps cases, we would expect the following: 1. Sequences from the

high-transmitting group should be more frequently detected upstream in transmission chains. 2. Introductions seeded into the high-transmitting group should result in larger and more diverse clades in the tree. 3. The internal nodes of the phylogeny should be predominantly composed by members of the high-transmitting group, while members of the low-transmitting group should primarily be found at terminal nodes, since less propagated transmission will cause the lineage to die out.

## Marshallese cases are enriched upstream in transmission chains

We developed a transmission metric to quantify whether Marshallese cases were enriched at the beginnings of successful transmission chains. We traverse the full genome divergence phylogeny (*Figure 2—figure supplement 4*) from root to tip. When we encounter a tip that lies on an internal node, we enumerate the number of tips that descend from its parent node. We then classify each tip in the phylogeny as either a 'basal tip' (i.e., there are tips detected downstream of that tip) or a 'terminal tip' (there are not tips detected downstream), and compare the proportion of basal and terminal tips among groups (*Figure 4a*, see Materials and methods for more details). Given our sampling proportion (110 sequences/889 total cases, ~12%), we do not expect to have captured true parent/child infection pairs. Rather, we expect to have preferentially sampled long, successful transmission chains within the state. This allowed us to test whether community membership, vaccination status, and age were associated with sustained transmission via logistic regression (see Materials and methods for details and statistical model). While those with unknown vaccination status were more likely to be basal in the tree than those with known up-to-date vaccination, the confidence interval could not exclude a null or positive association between vaccination status and basal/terminal status (*Table 1*). Having an age of at least 20 years predicted a mean lowered odds of being basal in the tree, but a wide range of effects is plausible given our sample. Resolving the precise effects of vaccination status and age would likely require a larger data set. However, we do find evidence for community status as a strong predictor for being basal on the phylogeny. Marshallese cases were significantly more likely to be basal than non-Marshallese cases (odds ratio = 3.2, p=0.00725, *Table 1*). While only 27% (14/52) of non-Marshallese tips were ancestral to downstream samples, 56% (32/57) of Marshallese tips were ancestral in lineages with sampled propagated transmission (*Supplementary file 1d*). These results suggest that community membership was a significant determinant of sustained transmission while controlling for vaccination status and age.

We evaluated the impact of vaccination status, age, and community membership on the probability that a sampled virus was basal in the tree. Coefficients represent the increase in the log odds of being basal in the tree for each given predictor variable while controlling for the others. Coefficients were exponentiated to produce odds ratios. We evaluated the impacts of having an unknown vaccination status, having a vaccination status that is not up-to-date, having an age of at least 20 years, and being Marshallese as binary predictor variables.

## Longer transmission chains are associated with community status

In the absence of recombination, closely linked infections will cluster together on the tree, while unrelated infections should fall disparately on the tree, forming multiple smaller clusters. We inferred the number of Washington-associated clades in the tree as a function of whether sampled infections came from Marshallese or non-Marshallese individuals. Using the full North American phylogeny, we removed all Washington sequences and separated them into viruses sampled from cases noted as Marshallese or non-Marshallese. Then, separately for each group, we added sequences back into the tree one by one, until all sequences for that group had been added. For each number of sequences, we performed 10 independent trials (see Materials and methods for complete details), and at each step, we enumerated the number of inferred Washington clusters in the phylogeny. For comparison, we also grouped tips by vaccination status and repeated this analysis.

For viruses sampled from non-Marshallese individuals, the number of inferred clusters increases linearly as tips are added to the tree (*Figure 4b*). This suggests that these infections are not closely related and are therefore not part of sustained transmission chains (*Figure 4b*). In contrast, the number of inferred clusters for Marshallese tips stabilizes after ~10 tips are added, even as almost 50 more sequences are added to the tree. This pattern likely arises because many Marshallese infections are part of the same long transmission chain, such that newly added tips nest within existing

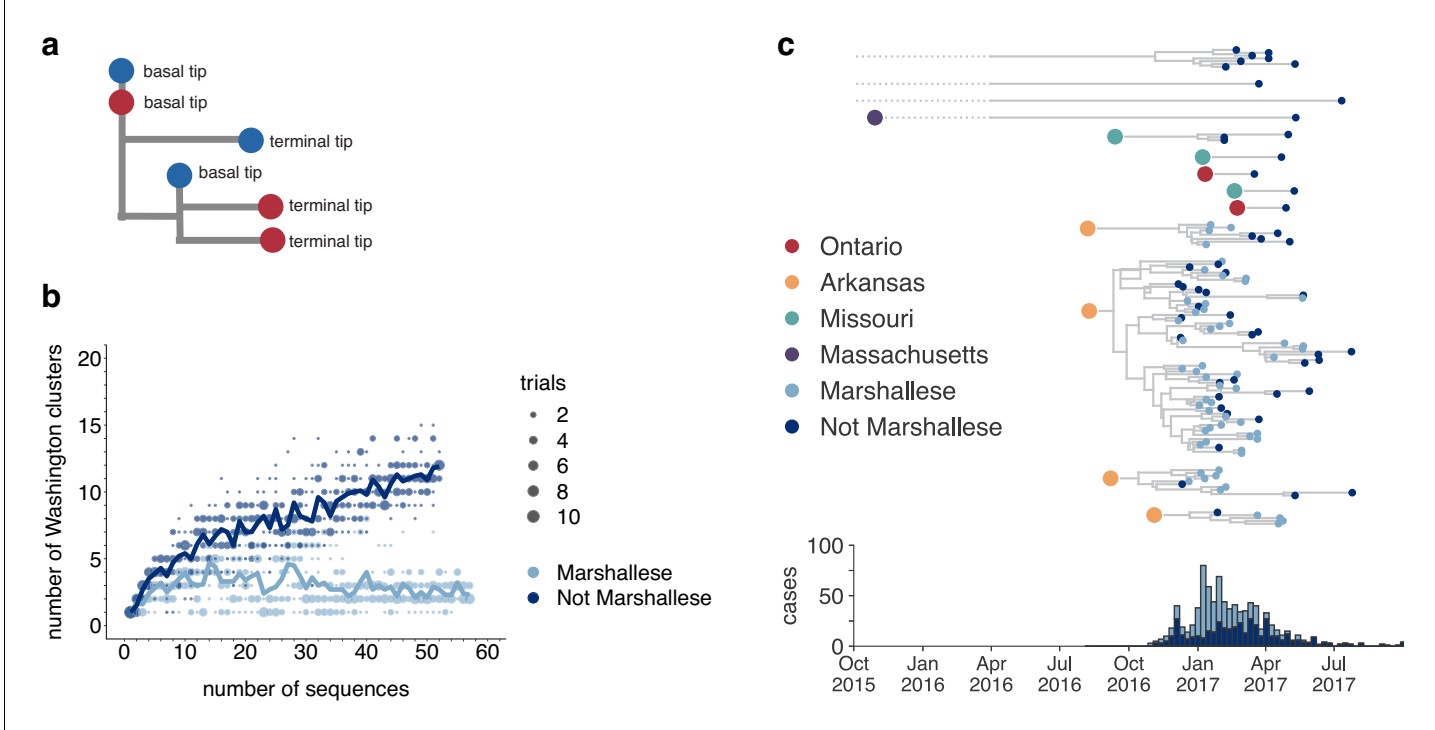

**Figure 4.** Marshallese individuals sustain longer transmission chains. (a) A schematic for quantifying tips that lie 'upstream' in transmission chains. For tips that lie on an internal node, meaning that they have a branch length separating them from their parent internal node of less than one mutation, we infer the number of child tips that descend from that tip's parental node. For each tip in the example tree, its classification as either a 'basal tip' or a 'terminal tip' is annotated alongside it. All tips that have a nonzero branch length are annotated as terminal tips. We can then compare whether sequences of particular groups (here, blue vs. red) are more likely to be basal or terminal via logistic regression. (b) We separated all Washington tips and classified them into Marshallese and not Marshallese. We then performed a rarefaction analysis and plotted the number of inferred Washington clusters (y-axis) as a function of the number of sequences included in the analysis (x-axis). Dark blue represents not Marshallese sequences, and light blue represents Marshallese sequences. Each dot represents the number of trials in which that number of clusters was inferred, and the solid line represents the mean across trials. (c) The exploded tree as shown in *Figure 3a* is shown, but tips are now colored by whether they represent Marshallese or non-Marshallese cases. For reference, the number of Washington cases (y-axis) is plotted over time (x-axis), where bar color represents whether those cases were Marshallese or not.

The online version of this article includes the following source data and figure supplement(s) for figure 4:

**Source data 1.** Rarefaction results for community status analysis shown in *Figure 4b*.

**Figure supplement 1.** Rarefaction results by vaccination status.

**Figure supplement 1—source data 1.** Rarefaction results for vaccination status analysis shown in *Figure 4—figure supplement 1*.

clusters. We do not observe similar differences among vaccination groups (*Figure 4—figure supplement 1*). These findings are consistent with distinct patterns of transmission among Marshallese versus non-Marshallese cases: transmission among Marshallese individuals resulted in a small number of large clusters, while transmission among non-Marshallese individuals are generally the result of disparate introductions that generate shorter transmission chains.

**Table 1.** Associations between basal tip position in the phylogeny and possible predictors of transmission.

| Predictor variable | Estimated coefficient (standard error) | Odds ratio (95% CI) | p-value |
|---|---|---|---|
| Not up-to-date | −0.76 (0.69) | 0.47 (0.11, 1.73) | 0.27 |
| Vaccination status unknown | 0.72 (0.77) | 2.04 (0.47, 10.15) | 0.35 |
| Age ≥20 years | −0.38 (0.51) | 0.69 (0.25, 1.86) | 0.46 |
| Community status | 1.21 (0.42) | 3.36 (1.49, 7.91) | 0.0042 |

We next separated each Washington introduction and colored each tip by community membership. Every introduction that was not seeded from Arkansas led to exclusively non-Marshallese infections, while introductions from Arkansas defined lineages that circulated for longer and were enriched with Marshallese tips (*Figure 4c*). The primary outbreak clade is particularly enriched, containing 43 Marshallese tips and 26 non-Marshallese tips, hinting that transmission chains are longer when Marshallese cases are present in a cluster.

## Mumps transmitted efficiently within the Marshallese community

Internal nodes on a phylogeny represent ancestors to subsequently sampled tips, while terminal nodes represent viral infections that did not give rise to sampled progeny. If the mumps outbreak was primarily sustained by transmission within one group, the backbone of the phylogeny and the majority of internal nodes should be inferred as circulating in that group. We selected the four introductions that contained both Marshallese and non-Marshallese tips (*Figure 4c*, the four Arkansas introductions), and reconstructed ancestral states along the phylogeny and migration/transmission rates between Marshallese and non-Marshallese groups using a structured coalescent model.

Of 88 internal nodes, 74 were inferred to circulate within the Marshallese community with posterior probability of at least 0.95 (*Figure 5a,b*). Movement of a lineage from the Marshallese deme into the non-Marshallese deme subsequently caused the lineage to die out quickly (*Figure 5a*, dark blue branches). This suggests that transmission was overwhelmingly maintained within the Marshallese community, and that infections seeded into the non-Marshallese community did not sustain prolonged transmission chains. We estimate substantially more transmission from Marshallese to non-Marshallese groups than the opposite: within the primary outbreak clade, we estimate 29 transmission events from Marshallese to non-Marshallese groups (95% HPD: 21, 37), and only 6 (95% HPD: 0, 14) from non-Marshallese to Marshallese groups (*Figure 5d*). This strongly suggests that transmission predominantly occurred in one direction: transmission events leading to non-Marshallese infections usually died out and did not typically re-seed circulation within the Marshallese community. These results hold true regardless of migration rate prior (*Figure 5—figure supplement 1*).

To ensure that our results were not driven by unequal sampling within the analyzed clades, we generated three data sets in which the number of Marshallese and non-Marshallese tips were subsampled to be equal. For each of these three subsampled data sets, we ran three independent chains under the same model described above. Chains converged for two of the three subsampled data sets. In the converged chains, we recover very similar tree topologies (*Figure 5—figure supplement 2a*) with equivalent phylogenetic reconstructions of lineage circulation within Marshallese and non-Marshallese demes. We also recovered maximum clade credibility trees in which the vast majority of the internal nodes are inferred to circulate within the Marshallese deme (*Figure 5—figure supplement 2a,b*), confirming that our findings are robust to sampling, consistent with the past observations of model performance (*De Maio et al., 2015*; *Dudas et al., 2018*; *Vaughan et al., 2014*).

The above structured coalescent model requires both groups to be present in each cluster, which meant that we had to exclude several small Washington introductions composed entirely of non-Marshallese tips (*Figure 4c*). To assess whether our findings would change if we analyzed all sequenced samples, we performed an additional analysis incorporating all Washington genotype G sequences in our data set and estimated a single tree using an approximate structured coalescent model (*Müller et al., 2018*). All Washington sequences were annotated as either Marshallese or not Marshallese. To provide a 'source' population for the extensive diversity among our disparate Washington introductions, we also specified a third, unsampled deme, for which migration was only allowed to proceed outward. As above, we inferred very few non-Marshallese internal nodes (*Figure 6* and *Figure 6—figure supplement 1*). All internal nodes in the primary outbreak group are inferred as Marshallese with high probability, while non-Marshallese cases are present as terminal nodes. We recovered support for a single non-Marshallese cluster, indicating limited sustained transmission in the non-Marshallese population.

Structured coalescent models infer the effective population size ($N_e$) for each group, which reflects the number of infections necessary to generate the observed genetic diversity. Differences in $N_e$ can result from different transmission rates or different numbers of infected individuals (*Volz, 2012*) and can therefore approximate differences in disease frequency between groups. While the total number of Marshallese and non-Marshallese cases reported through the public health

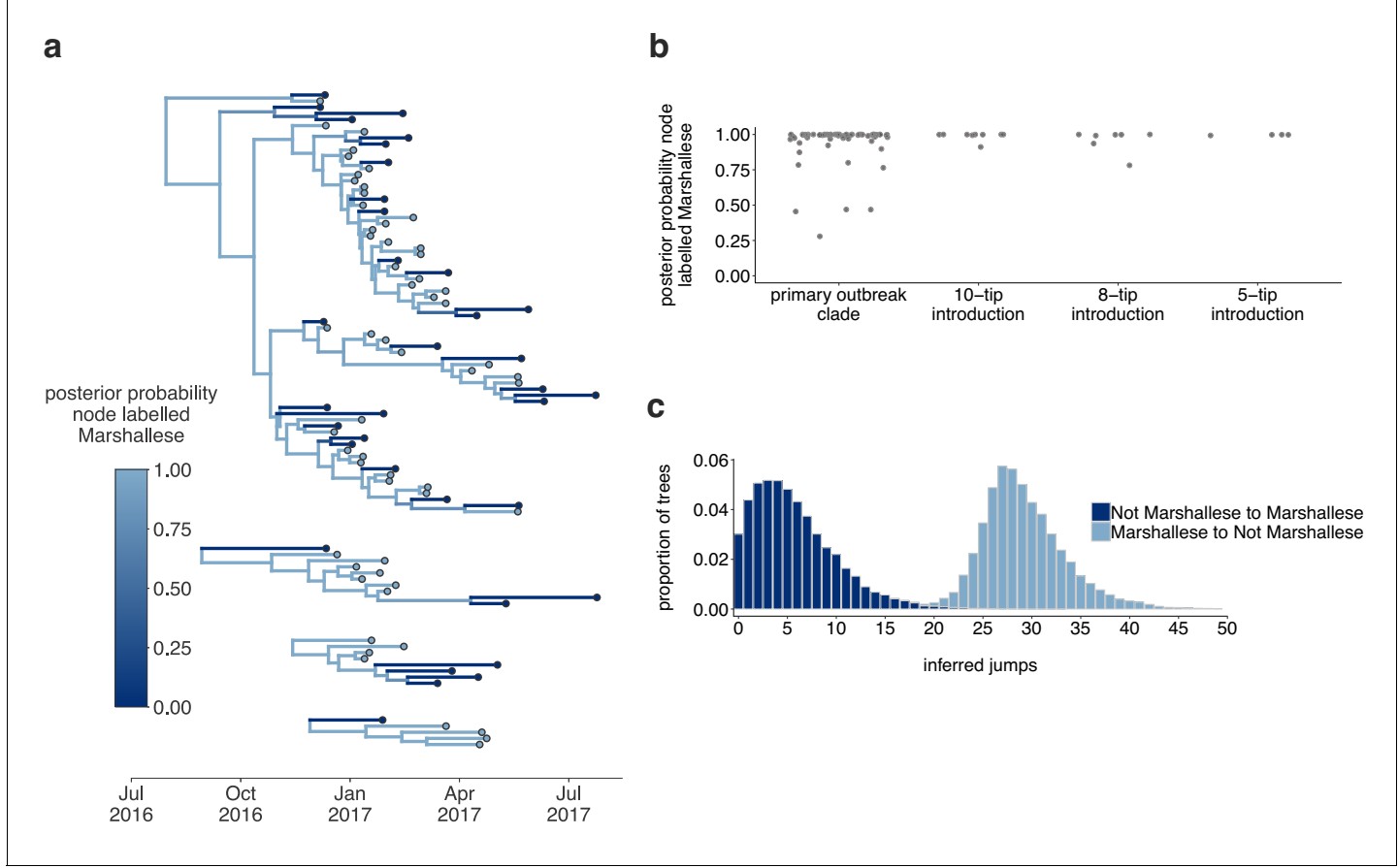

**Figure 5.** The Washington outbreak was sustained by transmission in the Marshallese community. (a) Using the four Washington clusters that had a mixture of Marshallese and non-Marshallese cases, we inferred phylogenies using a structured coalescent model. Each group of sequences shared a clock model, migration model, and substitution model, but each topology was inferred separately, allowing us to incorporate information from all four clusters into the migration estimation. For each cluster, the maximum clade credibility tree is shown, where the color of each internal node represents the posterior probability that the node is Marshallese. (b) For each internal node shown in panel (a), we plot the posterior probability of that node being Marshallese. Across all four clusters, 74 out of 88 internal nodes (84%) are inferred as Marshallese with a posterior probability of at least 0.95. (c) The posterior distribution of the number of 'jumps' or transmission events from Marshallese to not Marshallese (light blue) and not Marshallese to Marshallese (dark blue) inferred for the primary outbreak clade.

The online version of this article includes the following source data and figure supplement(s) for figure 5:

**Source data 1.** XML file to run structured coalescent analysis and combined output log and tree files with a migration rate prior of 1 (shown in *Figure 5*, identifiable metadata have been removed).

**Figure supplement 1.** Inferences are similar under a higher migration rate prior.

**Figure supplement 1—source data 1.** XML file to run structured coalescent analysis and combined output log and tree files with a migration rate prior of 10 (shown in *Figure 5—figure supplement 1*, identifiable metadata have been removed).

**Figure supplement 2.** Structured coalescent analyses are robust to sampling differences.

**Figure supplement 2—source data 1.** XML files and combined output files to run structured coalescent analysis where clades were subsampled to have equal numbers of Marshallese and non-Marshallese tips (shown in *Figure 5—figure supplement 2*, identifiable metadata have been removed).

surveillance system were similar (*Supplementary file 1b*), we estimate that $N_e$ for the non-Marshallese group is approximately three times higher than that of the Marshallese group. Assuming the same number of infected individuals in each group, lower $N_e$'s suggest higher transmission rates (*Volz, 2012*), suggesting more transmission within the Marshallese deme. Taken together, our results suggest that the outbreak was primarily sustained by transmission within the Marshallese community. While we do observe spillover into the non-Marshallese community, transmission was generally not as successful there, resulting in short, terminal transmission chains.

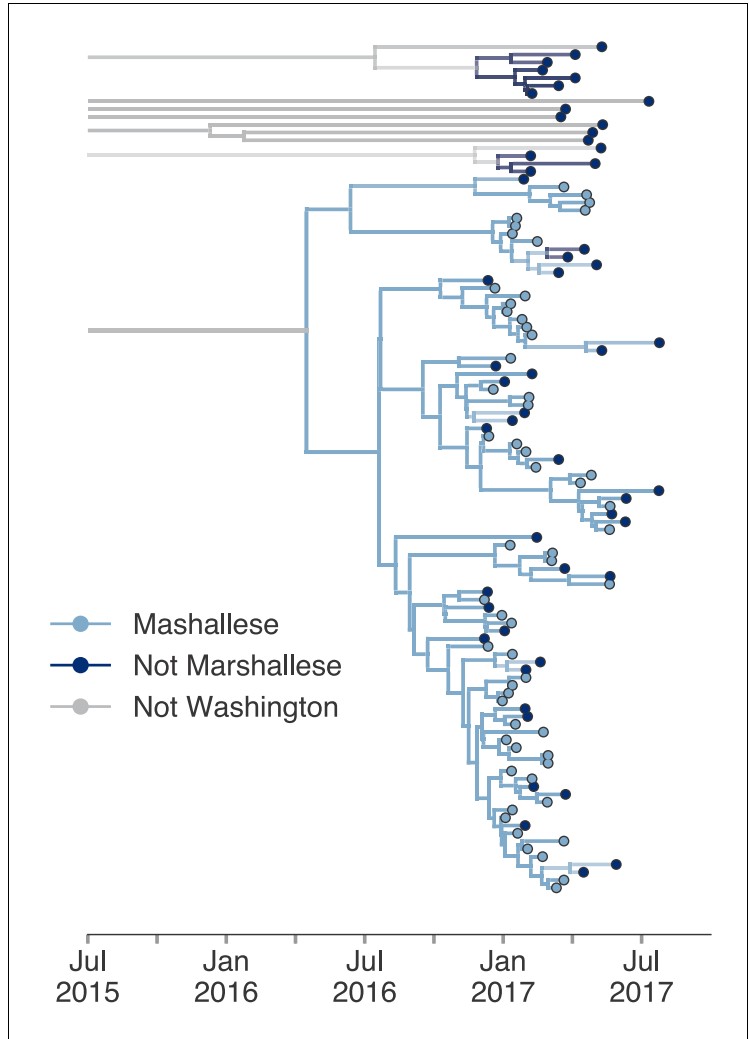

**Figure 6.** Including all Washington sequences recovers majority of transmission in Marshallese. To ensure that excluding non-Marshallese clusters did not skew our findings, we inferred a single tree using all Washington sequences. We performed a structured coalescent analysis specifying three groups: Marshallese, not Marshallese, and not Washington. Each internal node is colored by its most probable group, with its opacity specifying the posterior probability of being in that group (fully opaque being probability = 1, fully transparent being probability = 0).

The online version of this article includes the following source data and figure supplement(s) for figure 6:

**Source data 1.** XML file and output files to run structured coalescent analysis with unsampled 'ghost' deme shown in *Figure 6* (identifiable metadata have been removed).

**Figure supplement 1.** Posterior probabilities of internal node states.

## Viruses infecting individuals in different vaccination groups are genetically similar

Although only 9.7% of reported mumps cases in Washington were not up-to-date for mumps vaccination, infection of these individuals could have disproportionately impacted transmission in the state. Emergence of an antigenically novel strain of mumps could also allow infection of previously vaccinated individuals, and result in different virus lineages infecting individuals in different vaccination categories. We colored the tips of all Washington cases in our phylogeny to represent whether they were derived from individuals who were up-to-date, not up-to-date, or whose vaccination status was unknown. Mirroring overall vaccination coverage in Washington, the vast majority of samples in our data set were from up-to-date individuals. The not up-to-date individuals present in our data set

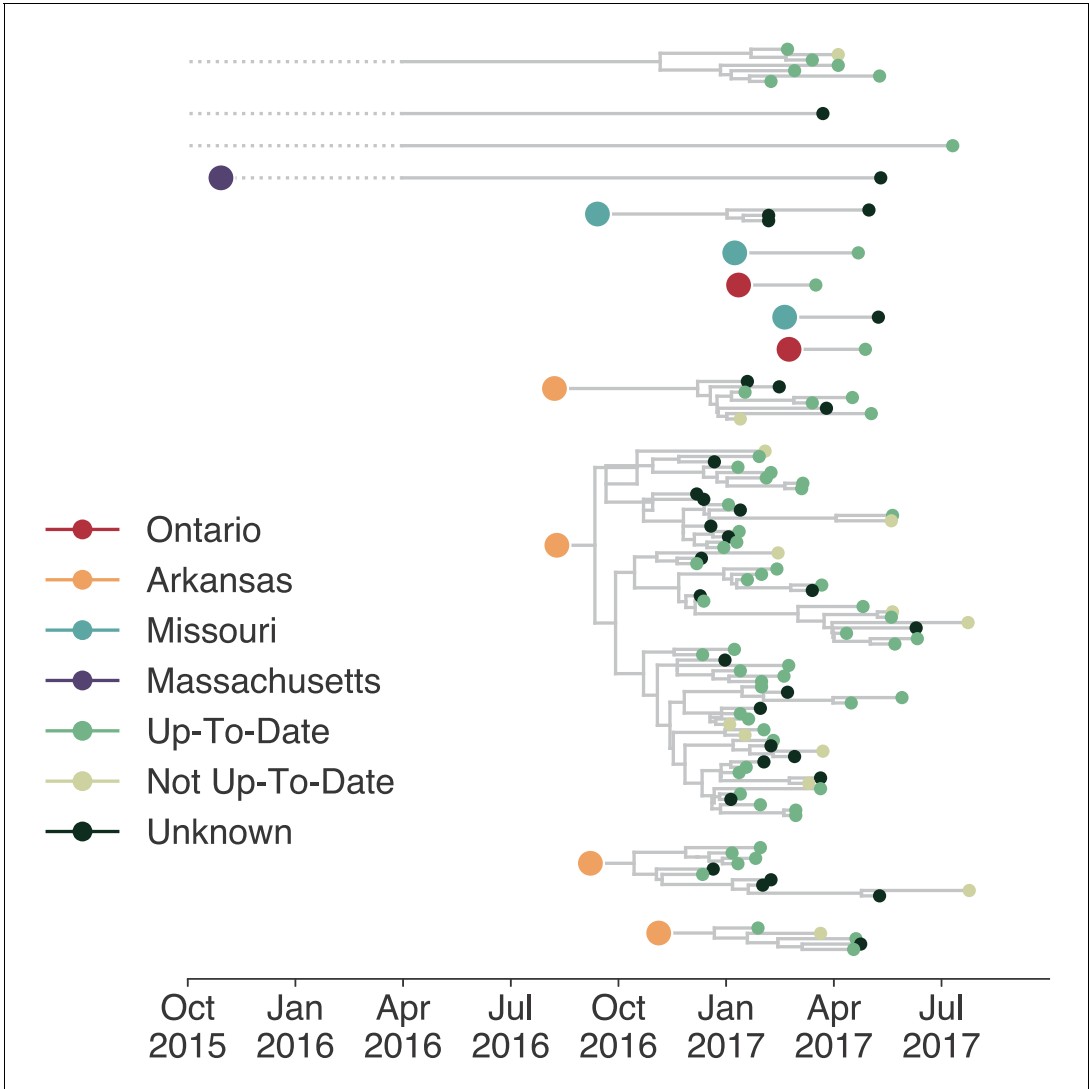

**Figure 7.** Individuals in different vaccination groups are infected by genetically similar viruses. The exploded tree as shown in *Figure 3a* is shown, but tips are now colored by whether they represent cases from individuals who are up-to-date for mumps vaccination, not up-to-date, or cases for which vaccination status was unknown. The color of the large dot represents the inferred geographic location from which the Washington introduction was seeded.

are dispersed throughout the phylogeny and do not cluster together (*Figure 7*), suggesting that there is no genetic difference between viruses infecting individuals with different vaccination statuses.

## Discussion

The resurgence of mumps in North America has ushered renewed attention toward understanding post-vaccine era mumps transmission. While many studies have used phylodynamic approaches to elucidate viral patterns of geographic spread (*Dudas et al., 2017*; *Gouma et al., 2016*; *Grubaugh et al., 2017*; *Stapleton et al., 2019*), using genomics to distinguish transmission patterns among epidemiologically distinct groups is novel. We employ a phylogenetic method (*Vaughan et al., 2014*) traditionally applied to geography that is robust to sampling bias (*Dudas et al., 2018*) to investigate drivers of mumps transmission in Washington. We show that the Washington State outbreak was fueled by approximately 13 independent introductions, primarily

from Arkansas, leading to multiple co-circulating transmission chains. Within Washington, transmission was more efficient within the Marshallese community. Marshallese individuals were more often sampled at the beginnings of transmission chains, contributed to longer transmission chains on average, and were overwhelmingly enriched on internal nodes within the phylogeny. We were unable to evaluate the precise effects of age and vaccination status on transmission in our outbreak. Future, larger studies will be necessary to disentangle the interplay between contact patterns, waning immunity, and vaccination status during mumps transmission. However, our data do suggest that social networks can be critical determinants of mumps transmission. Future work exploring how social and economic disparities may amplify respiratory disease transmission will be necessary for updating outbreak mitigation and prevention strategies. By combining detailed metadata, novel metrics of transmission in the tree, and robust controls for sampling, we provide a framework for investigating source-sink dynamics that is readily applicable to other viral pathogens.

Sampling bias presents a persistent problem for phylodynamic studies that can complicate inference of source-sink dynamics (*De Maio et al., 2015*; *Dudas et al., 2018*; *Frost et al., 2015*; *Kühnert et al., 2011*; *Lemey et al., 2020*; *Stack et al., 2010*). Sampling bias can arise from unequal case detection or from curating a data set that poorly represents the underlying outbreak. Washington State uses a passive surveillance system for mumps detection and case acquisition, which is known to result in underreporting. Because the WA Department of Health did not perform active mumps surveillance, it is difficult to assess whether different epidemiologic groups have different likelihoods of being sampled. Marshallese individuals are less likely to seek healthcare (*Towne et al., 2021*), which may have resulted in particularly high rates of underreporting in this group. If the number of cases within the Marshallese community were in fact higher than reported, this would increase the magnitude of the patterns we describe, making our estimates conservative. Given a distribution of cases, composing a data set for analysis also requires sampling decisions. Uniform sampling regimes in which sampling probability is equal across groups have been shown to perform well for source-sink inferences (*Hall et al., 2016*). By selecting sequences that matched the overall attributes of the outbreak, including a near 50:50 split between Marshallese and non-Marshallese cases, we adhere to this recommendation. We then specifically employed structured coalescent approaches which have been shown to be robust to sampling differences (*Dudas et al., 2018*; *Müller et al., 2018*; *Vaughan et al., 2014*), rather than using other common approaches that treat sampling intensity as informative of population size (*Lemey et al., 2009*). Within this framework, we further explore the possibility that unequal sampling within Washington clades could skew internal node reconstruction by forcing the sampling within each Washington clade to be equal between Marshallese and non-Marshallese tips. In doing so, differences within each clade must necessarily be driven by differences in transmission dynamics, rather than sampling. By combining careful sample selection with overlapping approaches to evaluate sampling bias, we were able to mitigate concerns that our source-sink reconstructions are driven by sampling artifacts.

Our results highlight the utility of genomic data to clarify epidemiologic hypotheses. While genomic data and epidemiologic investigation (including case interviews and contact follow up) suggested an Arkansas introduction as the Washington outbreak's primary origin, sequence data revealed repeated and ongoing introductions into Washington, similar to patterns observed in Massachusetts, and the Netherlands (*Gouma et al., 2016*; *Wohl et al., 2020*). We also find widespread geographic mixing across the phylogeny, consistent with investigations from the US (*Wohl et al., 2020*), Canada (*Stapleton et al., 2019*), and Europe (*Gavilán et al., 2018*; *Gouma et al., 2016*). Like others (*Gouma et al., 2016*; *Wohl et al., 2020*), we confirm that SH genotyping alone is insufficient for fine-grained resolution of geographic transmission patterns. While CDC guidelines currently recommend SH-based genotyping specifically for tracking transmission pathways (*Clemmons et al., 2020*), building public health capacity for full-genome sequencing may be more useful for resolving local mumps transmission patterns.

Our finding that most introductions sparked short transmission chains suggests that mumps did not transmit efficiently among the general Washington populace. We suspect that more diffuse contact patterns may help explain this. Mumps has historically caused outbreaks in communities with strong, interconnected contact patterns (*Barskey et al., 2012*; *Fields et al., 2019*; *Nelson et al., 2013*), and in dense housing environments (*Snijders et al., 2012*), highlighted most recently by outbreaks in US detention centers (*Lo et al., 2021*). In 2016, most outbreaks in the US were associated with university settings (*Albertson et al., 2016*; *Bonwitt et al., 2017*; *Donahue et al., 2017*;

*Golwalkar et al., 2018*; *Iowa Mumps Outbreak Response Team et al., 2018*; *Wohl et al., 2020*), including a separate, smaller outbreak in Washington State associated with Greek housing (*Bonwitt et al., 2017*). Outside of university settings, other outbreaks in 2016 were reported within close-knit ethnic communities (*Fields et al., 2019*; *Marx et al., 2018*). We speculate that while waning immunity may promote outbreaks by increasing susceptibility among young adults, outbreaks in younger age groups may be possible in sufficiently high-contact settings. Provision of an outbreak dose of mumps-containing vaccine to high-risk groups may therefore be especially effective for limiting mumps transmission in future outbreaks. Others have reported success in using outbreak dose mumps vaccinations to reduce mumps transmission on college campuses (*Cardemil et al., 2017*; *Iowa Mumps Outbreak Response Team et al., 2018*) and in the US army (*Arday et al., 1989*; *Eick et al., 2008*; *Green, 2006*; *Kelley et al., 1991*), and the CDC currently recommends providing outbreak vaccine doses to individuals with increased risk due to an outbreak (*Marlow et al., 2020*). Future work to quantify the interplay between contact rates and vaccine-induced immunity among different age and risk groups should be used to guide updated vaccine recommendations.

Recent research has focused on identifying groups at risk for mumps infection due to their age (*Lewnard and Grad, 2018*), with less attention to other factors that may make populations vulnerable. While a combination of waning immunity and dense housing settings make college campuses ideal for mumps outbreaks, the Washington and Arkansas outbreaks show that populations other than young adults are at risk. Soliciting feedback from the Marshallese community allowed us to contextualize our genomic results with the lived experience of individuals most heavily affected during the outbreak and to identify reasonable hypotheses for efficient transmission. Based on these interviews and previously published studies, we speculate that within the Marshallese community, a combination of factors likely led to a high force of infection. The following paragraph outlines contributing factors that were brought to light during our interviews with a collaborating community activist, along with corroborating citations from the literature. Each of these factors were specifically cited as important and directly stem from our interviews with her.

Multigenerational living is common in the Marshallese community (*Fields et al., 2019*), and Marshallese households tend to be larger on average (average household size = 5.28 (*Harris and Jones, 2005*), average household size for entire US populace = 2.52 *US Census Bureau, 2021*). Having more household contacts may have facilitated a greater number and higher intensity of interactions among individuals, allowing the force of infection to overcome pre-existing immunity. The Marshallese community is often described as close-knit, with frequent and close interactions among individuals, a strong sense of community, and a broader sense of family than the single-family unit typical of broader American culture (*Barker, 2012*; *Embassy of the Republic of the Marshall Islands to the United States of America, 2021*). Contacts within the community could therefore be more frequent or intense, which may facilitate transmission. It is also possible that infection intensity within the Marshallese community was exacerbated by low rates of insurance coverage and poor access to healthcare (*McElfish et al., 2017*; *Towne et al., 2021*), hesitancy to seek medical care (*Williams and Hampton, 2005*), and health disparities stemming from US occupation, nuclear testing, and exclusion from healthcare services. As part of reparations for US nuclear testing, the US signed the Compact of Free Association Treaty (COFA)(*Congress 108th United States, 2003*) with the Marshall Islands in 1989, permitting Marshallese residents to live and work in the US without visas. However, eligibility for Medicaid was revoked for COFA immigrants in 1996, and US-residing Marshallese remain economically disadvantaged and under-insured (*McElfish, 2016*; *McElfish et al., 2017*, *McElfish et al., 2015*). The passage of the Affordable Care Act (ACA) has not ameliorated these issues. Interviews with US-residing Marshallese note confusion among ACA staff regarding the legal status of COFA recipients, leading to drawn out enrollment processes that often leave individuals uninsured, frustrated (*McElfish et al., 2016*), and far less likely to access care (*Towne et al., 2021*). A study of healthcare-seeking behavior among patients with diabetes showed that while multiple factors contribute to foregone care in the US populace, 77% of surveyed Marshallese individuals reported recent forgeone care and lack of insurance was the primary reason (*Towne et al., 2021*). Marshallese trust in US medical institutions was seriously undermined by the unconsented use of Marshallese individuals for experiments on health impacts of nuclear exposure, with effects lingering today (*Barker, 2012*). Banked historical samples confirm uptake of radioactive materials in Marshallese inhabitants of affected Islands (*Simon et al., 2010*), but there has been limited published data on long-term health impacts of nuclear exposure, and significant concern remains within the

community (*Bordner et al., 2016*). Finally, when Marshallese individuals do access care, they report experiencing disdain from healthcare workers (*Duke, 2017*) and sub-optimal care (*McElfish et al., 2016*). Interviews with medical workers show that blame for poor Marshallese health outcomes is sometimes placed on host genetics or cultural practices (*Duke, 2017*), poor health literacy (*McElfish et al., 2018*), or choosing to delay care (*McElfish et al., 2018*), with less consideration given to how the economic and legal impacts of US occupation affect the health of Marshallese individuals. These factors compound, and Marshallese individuals report hesitation to seek medical care, even when sick (*McElfish et al., 2016*). Hesitancy to seek care could have contributed to mumps transmission if sick individuals were primarily cared for at home without knowledge of or the ability to implement community-isolation protocols.

Our findings highlight that social networks can be the primary risk factor for a respiratory virus outbreak, even when a vaccine is effective and widely used. This finding is especially pertinent as SARS-CoV2 continues to disproportionately impact populations who live and work in high-risk settings, including the Marshallese (*Center et al., 2020*; *McElfish et al., 2021*), and for whom vaccine licensure and distribution alone may not be a panacea. Future work should explore whether nuclear exposure has impacted Marshallese immune function and susceptibility to infectious disease. The passing of federal legislation remedying the exclusion of Marshallese individuals from Medicaid access (*Hirono, 2019*) in December 2020 marks an important step toward improving healthcare access. Future work to evaluate whether this change improves Marshallese access to healthcare and mitigates increased disease risk will be crucial follow-up. The findings of this paper demonstrate the importance of expanding our understanding of populations at risk for mumps re-emergence, so that rapid and comprehensive outbreak response strategies can be implemented to mitigate negative health impacts for all affected communities. Finally, future work to disentangle the complex interplay between healthcare access, social and economic disparity, and respiratory virus risk will be essential for mitigating health impacts of mumps and other respiratory viruses.

# Materials and methods

**Key resources table**

| Reagent type (species) or resource | Designation | Source or reference | Identifiers | Additional information |
|---|---|---|---|---|
| Biological sample (Mumps virus) | 110 buccal swabs from mumps positive patients in Washington | Washington State Department of Health | Sequences were deposited in Genbank under accessions MT859507-MT859672. Raw reads were deposited under SRA project number PRJNA641715 | Full metadata for each sequence is available in the manuscript in *Supplementary file 1a* |
| Biological sample (Mumps virus) | 56 buccal swabs from mumps positive patients from other US states | Wisconsin State Lab of Hygiene | Sequences were deposited in Genbank under accessions MT859507-MT859672. Raw reads were deposited under SRA project number PRJNA641715 | Full metadata for each sequence is available in the manuscript in *Supplementary file 1a* |
| Biological sample (Mumps virus) | Publicly available mumps genomes | NIAID Virus Pathogen Database and Analysis Resource (ViPR) (*Pickett et al., 2012*) | http://www.viprbrc.org/ | |
| Sequence-based reagent | mumps_1.5 kb primers | This paper | PCR primers | Full list of PCR primer sequences is available in the Materials and methods section under 'Viral RNA extraction, cDNA synthesis, and amplicon generation' |
| Commercial assay or kit | QiAmp Viral RNA Mini Kit | Qiagen, Valencia, CA, USA | Cat #: 52904 | |
| Commercial assay or kit | Protoscript II First strand synthesis kit | New England Biolabs, Ipswich MD, USA | Cat #: E6560L | |
| Commercial assay or kit | Q5 Hotstart DNA polymerase | New England Biolabs, Ipswich, MD, USA | Cat #: M0493L | |

*Continued on next page*

*Continued*

| Reagent type (species) or resource | Designation | Source or reference | Identifiers | Additional information |
|---|---|---|---|---|
| Commercial assay or kit | Ampure XP beads | Beckman Coulter | Cat #: A63881 | |
| Commercial assay or kit | Nextera XT DNA Library Prep Kit | Illumina, San Diego, CA, USA | Cat #: FC-131–1096 | |
| Software, algorithm | Bowtie2 | *Langmead and Salzberg, 2012* | http://bowtie-bio. sourceforge.net/ bowtie2/index.shtml | RRID:SCR_016368 |
| Software, algorithm | MAFFT | *Katoh et al., 2002* | https://mafft.cbrc.jp/ alignment/software/ | RRID:SCR_016368 |
| Software, algorithm | TreeTime | *Sagulenko et al., 2018* | https://github.com/ neherlab/treetime | |
| Software, algorithm | BEAST (versions 1.8.4 and 2.6.2) | *Drummond et al., 2012*, *Lemey et al., 2009*, *Bouckaert et al., 2019* | https://beast.community/ and https://www.beast2.org/ | RRID:SCR_010228 |
| Software, algorithm | IQTREE | *Nguyen et al., 2015* | http://www.iqtree.org | |
| Software, algorithm | Github repo with protocols for generating mumps sequences from buccal swabs | This paper | https://github.com/ blab/mumps-seq | This github repository contains documentation and protocols for all lab procedures and bioinformatics pipelines used to generate consensus genomes from mumps buccal swabs |
| Software, algorithm | Github repo with scripts used to analyze data and generate figures for this manuscript | This paper | https://github.com/ blab/mumps-wa-phylodynamics | This github repository contains all of the code used to generate figures and perform the analyses described in this manuscript. This repository also contains xml files used for input for BEAST analyses and alignments and tree files used to generate and plot phylogenetic trees |

## Data and code availability

All code used to analyze data, input files for BEAST, and all code used to generate figures for this manuscript are publicly available at https://github.com/blab/mumps-wa-phylodynamics; swh:1:rev: b8358a0d49d70670dbab9eeffa9972c277b3021b; *Moncla, 2021b*. Raw FASTQ files with human reads removed are available under SRA project number PRJNA641715. All protocols for generating sequence data as well as the consensus genomes are available at https://github.com/blab/mumps-seq; swh:1:rev:3309d1535804a71e6d9e7cc55295b6ea61bde730; *Moncla, 2021a*. Consensus genomes have also been deposited to Genbank under accessions MT859507-MT859672.

## Community feedback

In order to ensure that this study was faithful to the experience of the Marshallese community in Washington State, we sought paid consultation from a local Marshallese community health advocate. We conducted video and telephone interviews to directly address the impacts of mumps transmission on the Marshallese community, community healthcare goals and priorities, and the impacts of the mumps outbreak on stigmatization. This feedback informed what is being presented herein, provided crucial context for understanding mumps transmission, and allowed us to work with the community to determine how best to discuss Marshallese involvement in the outbreak.

## Mumps surveillance in Washington State

Mumps is a notifiable condition in Washington State. Therefore, per the Washington Administrative Code (WAC), as specified in WAC Chapter 246-101 (*Washington State Legislature, 2014*), health-care providers, healthcare facilities, and laboratories must report cases of mumps or possible mumps

to the local health jurisdiction (LHJ) of the patient's residence. LHJ staff initiate case investigations and facilitate optimal collection and testing of diagnostic specimens. Buccal swabs and urine are acceptable specimens for real-time reverse transcription polymerase chain reaction (qRT-PCR), a preferred diagnostic test for mumps. Most mumps rRT-PCR tests for Washington State residents are performed at the Washington State Public Health Laboratories, where all positive specimens are archived.

Individuals testing positive for mumps ribonucleic acid (RNA) by qRT-PCR are classified as confirmed mumps cases if they have a clinically compatible illness (i.e., an illness involving parotitis or other salivary gland swelling lasting at least 2 days, aseptic meningitis, encephalitis, hearing loss, orchitis, oophoritis, mastitis, or pancreatitis). During case investigations, case-patients or their proxies are interviewed. Information about demographics, illness characteristics, vaccination history, and potential for exposure to and transmission of mumps are solicited from each case-patient. In concordance with CDC guidelines (*Centers for Disease Control and Prevention, 2019b*), only vaccine doses for which there was written documentation with the date of vaccine receipt were considered valid. Individuals for whom such documentation could not be provided were classified as having an unknown vaccination status. For individuals with documented vaccine doses, they were further characterized as up-to-date or not up-to-date based on their age. The Washington State Department of Health (DOH) receives, organizes, performs quality control on, and analyzes data from, LHJ case reports and supports investigations upon request.

## Sample collection and IRB approval

This study was approved by the Fred Hutchinson Cancer Research Center (FHCRC) Institutional Review Board (IR File #: 6007–944) and by the Washington State Institutional Review Board, and classified as not involving human subjects. Samples were selected for sequencing to maximize temporal and epidemiologic breadth and to ensure successful sequencing. As such, samples were chosen based on the date of sample collection, the PCR cycle threshold (Ct), case vaccination status, and community status (Marshallese or non-Marshallese). Samples were selected for sequencing in two batches. In the first, samples were selected based on covering a wide geographic range within Washington, a full range of dates covering the outbreak, and having a Ct value <36. This initial sampling regime resulted in a sample set skewed slightly toward samples from Marshallese individuals. To ensure that the proportion of samples in our data closely matched the distribution of cases in the outbreak, we then selected a second batch of samples using the same criteria as above, but excluded samples from Marshallese individuals. We then randomly sampled an additional 30 samples from non-Marshallese individuals. This sampling regime resulted in a data set that closely mirrors the distribution of metadata categories in the outbreak overall. All metadata, including case vaccination status, were transferred from WA DOH to FHCRC in a de-identified form.

We also sequenced an additional set of 56 samples collected in Wisconsin, Ohio, Missouri, Alabama, and North Carolina provided by the Wisconsin State Laboratory of Hygiene. Ten of these samples were collected in Wisconsin during the 2006/2007 Midwestern college campus outbreaks, six samples were collected in 2014, and the rest were collected between 2016 and 2018. For these samples, we received metadata describing sample Ct value and date of collection. All metadata were received by FHCRC in de-identified form.

## Viral RNA extraction, cDNA synthesis, and amplicon generation

Viral RNA was extracted from buccal swabs using either the QiAmp Viral RNA Mini Kit (Qiagen, Valencia, CA, USA) or the Roche MagNA Pure 96 DNA and viral NA small volume kit (Roche, Basel, Switzerland). For samples extracted with the QiAmp Viral RNA Mini Kit, 500 µl of buccal swab fluid was spun at 5000 × g for 5 min at 4℃ to pellet host cells. The supernatant was then removed and centrifuged at 14,000 rpm for 90 min at 4℃ to pellet virions. Excess fluid was discarded, and the pelletted virions were resuspended in 150–200 µl of fluid. Resuspended viral particles were then used as input to the QiAmp Viral RNA Mini Kit (Qiagen, Valencia, CA, USA), following manufacturer's instructions, and eluting in 30 µl of buffer AVE. For extraction with the MagNA Pure, we followed manufacturer's instructions. cDNA was generated with the Protoscript II First strand synthesis kit (New England Biolabs, Ipswich MD, USA), using 8 µl of vRNA as input and priming with 2 µl of random hexamers. vRNA and primers were incubated at 65℃ for 5 min. Following this incubation, 10 µl

of Protoscript II reaction mix (2×) and 2 µl of Protoscript II enzyme mix (10×) were added to each reaction and incubated at 25℃ for 5 min, then 42℃ for 1 hr, followed by a final inactivation step at 80℃ for 5 min. To amplify the full mumps genomes, we used Primal Scheme (http://primal.zibrapro-ject.org/) to design overlapping,~1500 base pair amplicons spanning the entirety of the mumps virus genome, where each tiled set of primes overlapped by ~100 base pairs. Primers are listed below.

| Primer | Primer sequence | Forward/reverse | Primer pool |
|---|---|---|---|
| mumps_1.5 kb_1F | ACCAAGGGGAAAATGAAGATGGG | Forward | pool 1 |
| mumps_1.5 kb_1R | TAACGGCTGTGCTCTAAAGTCAT | Reverse | pool 1 |
| mumps_1.5 kb_2_F | TTGTTGACAGGCTTGCAAGAGG | Forward | pool 2 |
| mumps_1.5 kb_2_R | TTGTTCAAGATGTTGCAGGCGA | Reverse | pool 2 |
| mumps_1.5 kb_3_F | TGCAACCCCATATGCTCACCTA | Forward | pool 1 |
| mumps_1.5 kb_3_R | AGTTTGTTCCTGCCTTTGCACA | Reverse | pool 1 |
| mumps_1.5 kb_4_F | AGTGAGAGCAGTTCAGATGGAAGT | Forward | pool 2 |
| mumps_1.5 kb_4_R | CCCTCCATTAGACCGGCACTTA | Reverse | pool 2 |
| mumps_1.5 kb_5_F | AACAACAGTGTTCCAGCCACAA | Forward | pool 1 |
| mumps_1.5 kb_5_R | GGTGGCACTGTCCGATATTGTG | Reverse | pool 1 |
| mumps_1.5 kb_6_F | TGCCGTTCAATCATGAGACATAAAGA | Forward | pool 2 |
| mumps_1.5 kb_6_R | CGTAGAGGAGTTCATACGGCCA | Reverse | pool 2 |
| mumps_1.5 kb_7_F | TGTCTGTGCCTGGAATCAGATCT | Forward | pool 1 |
| mumps_1.5 kb_7_R | CGTCCTTCCAACATATCAGTGACC | Reverse | pool 1 |
| mumps_1.5 kb_8_F | CCAAAAGACAGGTGAGTTAACAGATTT | Forward | pool 2 |
| mumps_1.5 kb_8_R | ACGAGCAAAGGGGATGATGACT | Reverse | pool 2 |
| mumps_1.5 kb_9_F | TTTGGCACACTCCGGTTCAAAT | Forward | pool 1 |
| mumps_1.5 kb_9_R | TGACAATGGTCTCACCTCCAGT | Reverse | pool 1 |
| mumps_1.5 kb_10_F | ACTCGCACAGTATCTATTAGATCGTGA | Forward | pool 2 |
| mumps_1.5 kb_10_R | GCCCAGCCAGAGTAAACAAACA | Reverse | pool 2 |
| mumps_1.5 kb_11_F | GCCAAGCAGATGGTAAACAGCA | Forward | pool 1 |
| mumps_1.5 kb_11_R | GGCTCTCTCCAACATGCTGTTC | Reverse | pool 1 |
| mumps_1.5 kb_12_F | GCGGGGCCTCTATGTCACTTAT | Forward | pool 2 |
| mumps_1.5 kb_12_R | CCAAGGGGAGAAAGTAAAATCAAT | Reverse | pool 2 |

Primers were pooled into two pools as follows: the first contained primer pairs 1, 3, 5, 7, 9, and 11, all pooled at 10 µM. The second pool contained primer pairs 2, 4, 6, 8, 10, and 12. All primers in pool two were pooled at 10 µM, except for primer pair 4, which was added at a 20 µM concentration.

PCR was performed with the Q5 Hotstart DNA polymerase (New England Biolabs, Ipswich, MD, USA), using 11.75 µl of nuclease-free water, 5 µl of Q5 reaction buffer, 0.5 µl of 10 mM dNTPs, 0.25 µl, 2.5 of pooled primers, and 5 µl of cDNA. Amplicons were generated with the following PCR cycling conditions: 98℃ for 30 s, followed by 30 cycles of: 98℃ for 15 s, then 67℃ for 5 min. Cycling was concluded with a 10℃ hold. PCR products were run on a 1% agarose gel, and bands were cut out and purified using the QiAquick gel extraction kit (Qiagen, Valencia, CA, USA), following the manufacturer's protocol. All optional steps were performed, and the final product was eluted in 30 µl of buffer EB. For samples extracted on the MagNA Pure, amplicons were cleaned using a 1× bead cleanup with Ampure XP beads. Final cleaned amplicons were quantified using the Qubit dsDNA HS Assay kit (Thermo Fisher, Waltham, MA, USA).

## Library preparation and sequencing

For each sample, pool 1 and pool 2 amplicons were combined in equimolar concentrations to a total of 0.5 ng in 2.5 µl. Libraries were prepared using the Nextera XT DNA Library Prep Kit (Illumina, San Diego, CA, USA), following manufacturer's instructions, but with reagent volumes halved for each

step, for the majority of samples in our data set. For samples processed in our last sequencing run, several samples had higher Ct values. We therefore chose to process these samples using the standard 1× reagent volumes for the library preparation step. All libraries were purified using Ampure XP beads (Beckman Coulter, Brea, CA, USA), using a 0.6× cleanup, a 1× cleanup, and a final 0.7× cleanup. At each step, beads were washed twice with 200–400 µl of 70% ethanol. The final product was eluted off the beads with 10 µl of buffer EB. Tagmentation products were quantified with the Qubit dsDNA HS Assay kit (Thermo Fisher, Waltham, MA, USA), and run on a Tapestation with the TapeStation HighSense D5K assay (Agilent, Santa Clara, CA, USA) to determine the average fragment length. All but eight samples and negatives were pooled together in 6 nM libraries and run on 300 bp × 300 bp v3 kits on the Illumina MiSeq, with a 1% spike-in of PhiX. The remaining eight samples (MuVs/Washington.USA/1.17/FH77[G], MuVs/Washington.USA/12.17/FH78[G], MuVs/Washington.USA/16.17/FH79[G], MuVs/Washington.USA/19.17/FH80[G], MuVs/Washington.USA/20.17/FH81 [G], MuVs/Washington.USA/20.17/FH82[G], MuVs/Washington.USA/29.17/FH83[G], and MuVs/Washington.USA/2.17/FH84[G]) were pooled to a 1.2 nM library, and run as a 50 pM library with 2% PhiX on the Illumina iSeq, with a 151 bp × 151 bp v3 kit.

## Negative controls

A negative control (nuclease-free water) was run for each viral RNA extraction, reverse transcription reaction, and for each pool for each PCR reaction. These negative controls were carried through the library preparation process and sequenced alongside actual samples. Any samples whose negative controls from any step in the process resulted in >10× mumps genome coverage were re-extracted and sequenced.

## Bioinformatic processing of sequencing reads

Human reads were removed from raw FASTQ files by mapping to the human reference genome GRCH38 with bowtie2 (*Langmead and Salzberg, 2012*)(RRID: SCR_016368) version 2.3.2 (http://bowtie-bio.sourceforge.net/bowtie2/index.shtml). Reads that did not map to the human genome were output to separate FASTQ files and used for all subsequent analyses. Illumina data was analyzed using the pipeline described in detail at https://github.com/lmoncla/illumina_pipeline. Briefly, raw FASTQ files were trimmed using Trimmomatic (*Bolger et al., 2014*) (http://www.usadellab.org/cms/?page=trimmomatic), trimming in sliding windows of 5 base pairs and requiring a minimum Q-score of 30. Reads that were trimmed to a length of <100 base pairs were discarded. Trimming was performed with the following command: java -jar Trimmomatic-0.36/trimmomatic-0.36.jar SE input.fastq output.fastq SLIDINGWINDOW:5:30 MINLEN:100. Trimmed reads were mapped to a consensus sequence from Massachusetts (Genbank accession: MF965301) using bowtie2 (*Langmead and Salzberg, 2012*) version 2.3.2 (http://bowtie-bio.sourceforge.net/bowtie2/index.shtml), using the following command: bowtie2 -x reference_sequence.fasta -U read1.trimmed.fastq,read2.trimmed.fastq -S output.sam –`local`. We selected this Massachusetts sequence as an initial reference sequence because at the time, it represented one of the only available genomes of a genotype G mumps virus that had been sampled during a US outbreak in 2016. Mapped reads were imported into Geneious (https://www.geneious.com/) for visual inspection and consensus calling. To avoid issues with mapping to an improper reference sequence, we then remapped each sample's trimmed FASTQ files to its own consensus sequence. These bam files were again manually inspected in Geneious, and a final consensus sequence was called, with nucleotide sites with <20× coverage output as an ambiguous nucleotide ('N'). All genomes with >50% Ns were discarded. In total, we generated 140 genomes with at least 80% non-N bases, and 26 genomes with 50–80% non-N bases. Our median completeness (percent of bases that are not Ns) across the data set is 90%. All genomes used in these analyses are available at https://github.com/blab/mumps-seq/tree/master/data.

## Data set curation and maximum likelihood divergence tree generation

We downloaded all currently available (as of June 2020), complete mumps genomes from North America, and separately from any country in the world, from the NIAID Virus Pathogen Database and Analysis Resource (ViPR) (*Pickett et al., 2012*) through http://www.viprbrc.org/. We also obtained mumps genomes from British Columbia, Ontario, and Arkansas. We obtained written

permission from sequence authors for any sequence that had not previously been published on. In total, this data set includes 437 full mumps genomes from North America. Sequences and metadata were cleaned and organized using fauna, a database system that is part of the Nextstrain platform. Sequences were processed using Nextstrain's augur software (*Hadfield et al., 2018*), and filtered to include only those with at least 8000 bases and were sampled in North America in 2006 or later. Genomes were aligned with MAFFT (*Katoh et al., 2002*)(RRID: SCR_016368) and trimmed to the reference sequence (MuV/Gabon/13/2[G], GenBank accession: KM597072). We inferred a maximum likelihood phylogeny using IQTREE (*Nguyen et al., 2015*) with a GTR nucleotide substitution model, and inferred a molecular clock and temporally resolved phylogeny using TreeTime (*Sagulenko et al., 2018*). Sequences with an estimated clock rate that deviated from the other sequences by >4 times the interquartile distance were removed from subsequent analysis. We inferred the root-to-tip distance with TempEst version 1.5.1 (*Rambaut et al., 2016*) with the best fitting root by the heuristic residual mean squared function. Trees were output in JSON format and are available at https://github.com/blab/mumps-wa-phylodynamics/blob/master/auspice.

## Phylogenetic analysis of full North American mumps genomes

Using the same set of genome sequences used for divergence tree estimation, we aligned sequences with MAFFT and inferred time-resolved phylogenies in BEAST version 1.8.4 (*Drummond et al., 2012*)(RRID: SCR_010228). We used a skygrid population size prior with 100 bins, and a skygrid cut-off of 25 years, allowing us to estimate four population sizes each year. We used an HKY nucleotide substitution model with four gamma rate categories, and a strict clock with a CTMC prior. We used a discrete trait model (*Lemey et al., 2009*) and estimated migration rates using BSSVS and ancestral states with 27 geographic locations. Here, 'state' refers to the inferred ancestral identity of an internal node, where the inferred identity could be any of the 27 geographic locations (US states and Canadian provinces) in the data set. For the prior on non-zero rates for BSSVS, we specified an exponential distribution with a mean of 26. As a prior on each pairwise migration rate, we used an exponential distribution with mean 1. All other priors were left at default values. We ran this analysis for 100 million steps, sampling every 10,000, and removed the first 10% of sampled states as burn-in. A maximum clade credibility tree was summarized with TreeAnnotator, using the mean heights option. All tree plotting was performed with baltic (https://github.com/evogytis/baltic). Input XML files and output results are available at https://github.com/blab/mumps-wa-phylodynamics/tree/master/phylogeography.

## Quantifying transmission in divergence trees using basal and terminal tips: formulation and rationale

To determine whether specific groups were more likely to be part of sustained, serially sampled transmission chains, we developed a statistic to quantify transmission in the tree. Our aim was to develop a heuristic method that would capture patterns similar to those captured by more complex structured coalescent models. In a population with high rates of transmission and high sampling intensity, it is possible that sampled individuals may represent true ancestors to subsequent infections (*Gavryushkina et al., 2014*). While this is theoretically possible in our data set, we expect this to be rare. Because viruses accumulate mutations at a constant rate over time, a tip's branch length should correlate with its position along the underlying transmission chain, that is, a short branch length should indicate that the tip is closer to the true ancestral infection than a longer branch. Plotting the number of mutations on each branch vs. its estimated branch length in time units confirms that mutations and time-calibrated branch length are correlated (*Figure 8*). This suggests that on average, branches with fewer mutations also tend to represent shorter periods of time. For our data set, most tips with an estimated branch length within the mumps serial interval of ~18 days *Vink et al., 2014* have 0 mutations (*Figure 8*, circles below dashed line). In a population with high rates of transmission in which true ancestral infections are not directly sampled, we therefore expect that tips that are genetically closer to the ancestral node should be closer to the true ancestral infection than tips that are genetically dissimilar.

Either time or genetic divergence could be used to categorize how close tips are to their parental node. Here, we have opted to use divergence for two main reasons. Mumps has a relatively slow substitution rate and a long serial interval, resulting in stacks of identical genomes at several points

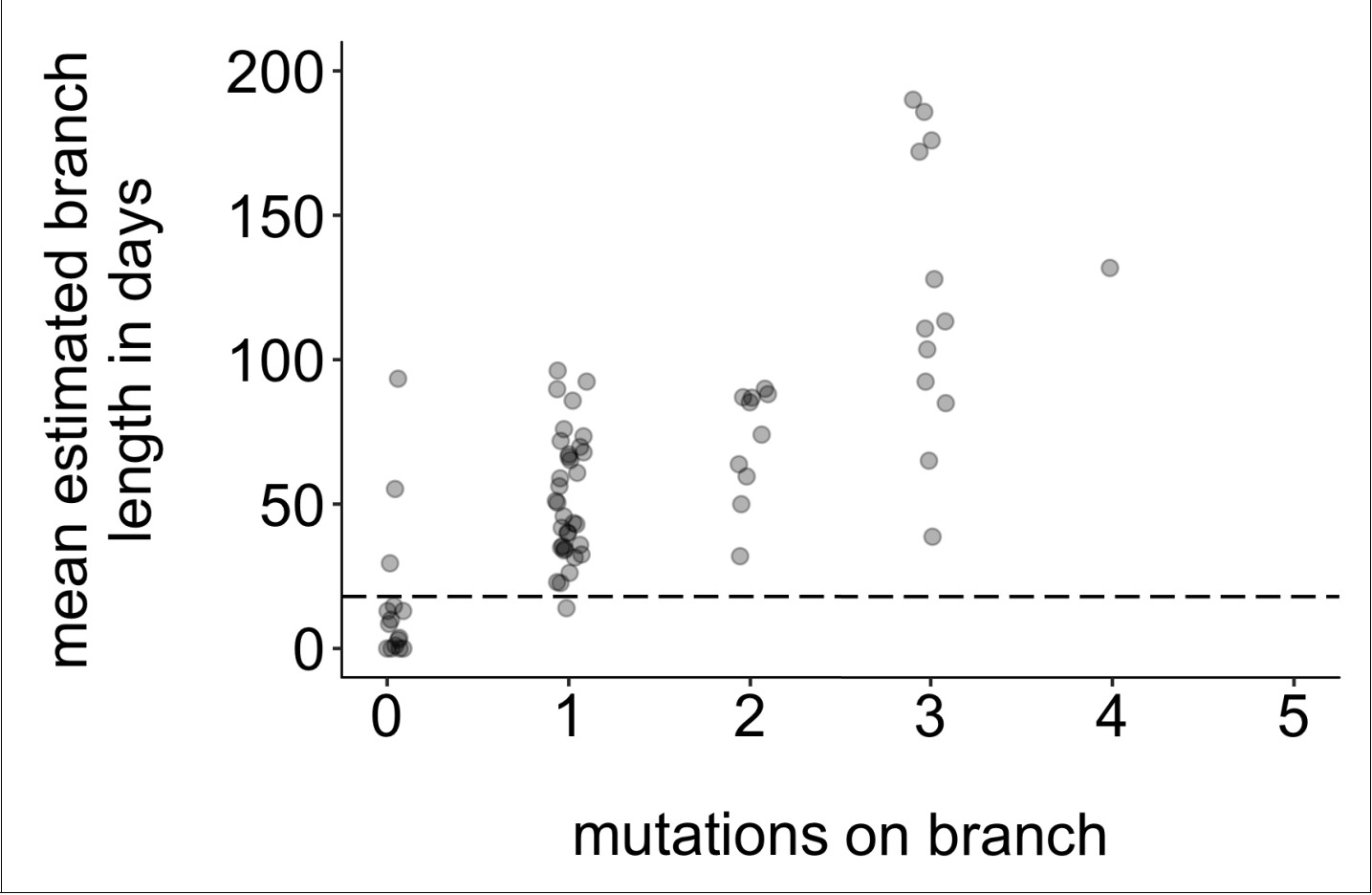

**Figure 8.** Mutations vs. estimated branch length in days. For each Washington tip in the full North American phylogeny with an estimated branch length in time units of ≤1 year, we show the number of mutations on that branch on the x-axis vs. the mean estimated branch length in days on the y-axis. The dashed line at 18 represents the mumps serial interval.

in the divergence phylogeny (*Figure 2—figure supplement 4*). In time-resolved phylogenies, branching among identical genomes is resolved by sampling date, and the x-coordinate of the internal node is inferred based on time information. Because the internal node location is based not on actual genetic information, there is variability in the estimated placement of the internal node on the tree, resulting in branch lengths that may vary among realizations of the tree. This is reflected in the 95% confidence interval of internal node dates. Plotting the estimated branch length from each Washington tip to its internal node and incorporating the 95% confidence interval of the internal node date show that a wide range of branch lengths are plausible for most tips (*Figure 9*). This complicates setting a simple branch length threshold based on the serial interval. In contrast, divergence trees have branch lengths expressed in the number of mutations arising along on that branch, which is intrinsic to the sequences themselves. Second, higher transmission in one group could result in shorter serial intervals within that group, which complicates defining a branch length cutoff based on serial interval. For these reasons, we have opted to use genetic divergence as our metric of 'closeness' to avoid arbitrary time cutoffs and issues of uncertainty in timetree internal node placements.

Given a divergence tree, we next categorized tips by how close they are to their parental node. To maximize the similarity between the tip and its inferred ancestral node, we classified tips as 'close' to their ancestral node if no mutations occurred on the branch leading to that tip, that is, the branch length was less than 1 divided by the alignment length. This cutoff can be set to the lower bound of the tree software (for IQTree, this cutoff is ~<$1 \times 10^{-16}$) with the same results. Using the JSON for the North American full genome mumps tree output from the Nextstrain pipeline (shown in *Figure 2—figure supplement 2*; *Hadfield et al., 2018*), we traversed the tree from root to tip.

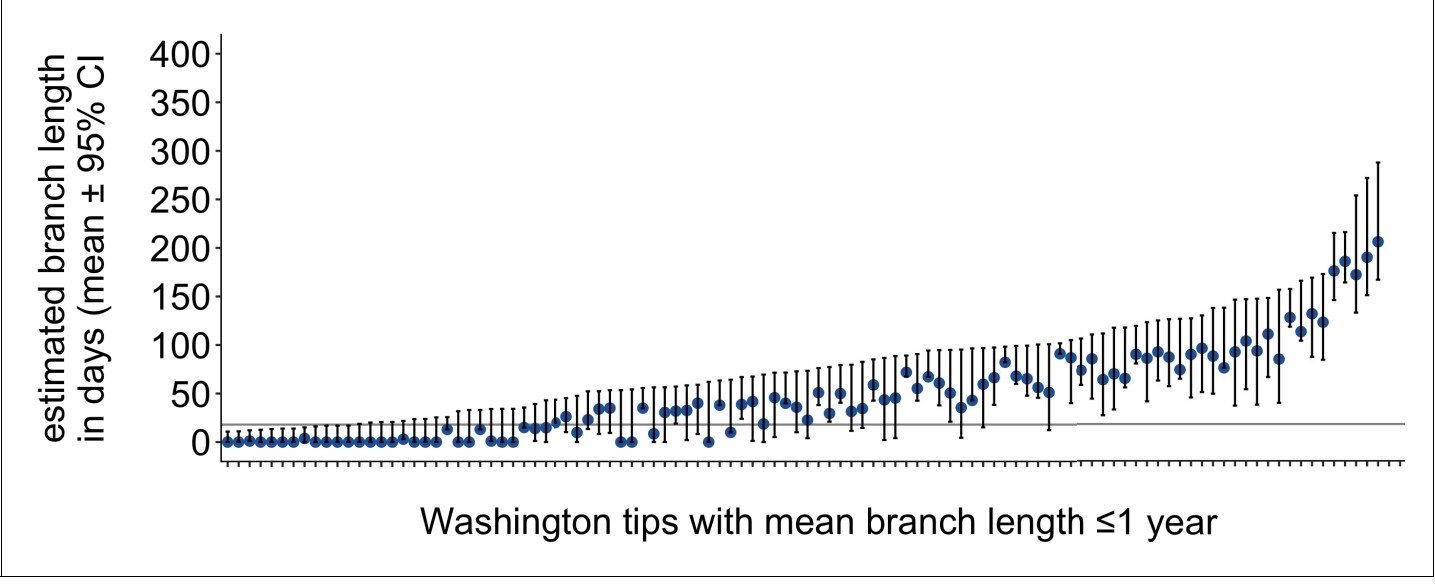

**Figure 9.** Washington tip branch lengths in days. For each Washington tip in the full North American phylogeny with an estimated branch length in time units of ≤1 year, we show the estimated branch length in days with the 95% confidence interval. The solid line at 18 represents the mump serial interval. For most tips, the estimated branch length is variable, depending on the placement of its parental internal node. This variability in internal node placement complicates setting a clear threshold for branch lengths based on time.

We collapsed very small branches (branches with no mutations) to obtain polytomies, and then classified tips as either 'basal' (i.e., there were tips in the tree that descended from that internal node) or 'terminal' (meaning that no sampled tips descended from that branch). Here, we define a 'descendant' tip as a tip that occurs in any downstream portion of the tree, that is, it falls within the same lineage but to the right of the parent tip. A diagram of what we classify as basal vs. terminal tips is shown in *Figure 4a*.

We expect that requiring branches to have 0 mutations should be robust regardless of mutation rate and serial interval, because a branch length of 0 will always be the closest in sequence to the true ancestor. However, variation in the substitution rate will impact the power of the analysis for detecting associations. Because mumps has a low substitution rate, some internal nodes contain stacks of identical genomes that cannot be ordered in terms of their placement along the underlying transmission chain. Instead, we treat each of these nodes as basal with equal probability of being upstream in the transmission chain. A higher substitution rate would jitter these polytomies and increase resolution, while a lower substitution rate would further reduce power. Our application is therefore conservative, but likely underpowered. Future work will be necessary to define the precise interaction between mutation rate, serial interval, sampling intensity, and effect sizes in determining the power of this test.

## Regression model for quantifying a tip's probability of being basal

For each Washington tip in the tree, we classified it as either being basal (coded as a 1) or being terminal (coded as 0). For each tip, we coded its corresponding age, vaccination status, and community membership as a predictor variable input into a logistic regression model. We coded these attributes as follows: For community membership, non-Marshallese tips were coded as 0 and Marshallese tips were coded as 1. For age, we split our data set into adults and children, with individuals aged <20 coded with a 0 and ≥20 coded with a 1. In our data set, there were three classifications for vaccination status: up-to-date, not up-to-date, and unknown vaccination status. According to the Advisory Committee on Immunization Practices (ACIP) (*McLean, 2013*), individuals aged 5–18 had to have received both recommended doses of mumps-containing vaccine, children aged 15 months to 5 years required one dose of mumps-containing vaccine, and adults over 18 had to have received at least one dose of mumps-containing vaccine to be classified as up-to-date for mumps vaccination.

Individuals under 15 months are considered up-to-date without any doses of mumps-containing vaccine. Not up-to-date individuals are those with a known vaccination status who did not qualify under criteria to be classified as up-to-date. Individuals who could not provide documentation regarding their MMR vaccination history were considered to have 'unknown' vaccination status. Individuals with 'known' vaccination status could either be fully up-to-date, undervaccinated, or unvaccinated. To ensure that we measured the effect of vaccination among individuals who knew their vaccination status, we coded vaccination information using two dummy variables in our logistic regression, one signifying whether vaccination status was known or not, and one indicating whether vaccination was up-to-date or not. We then fit a logistic regression model to this data using the glm package in R (https://www.rdocumentation.org/packages/stats/versions/3.6.2/topics/glm), specifying a binomial model as:

$$\Pr(\text{being basal}) \sim \beta_0 + \beta_1 x_1 + \beta_2 x_2 + \beta_3 x_3 + \beta_4 x_4,$$

where $x_1$ represents 0 or one value for member of Marshallese community (not Marshallese coded as a 0, Marshallese coded as a 1), $x_2$ represents a 0 or one value for age, where individuals were classified as adults ($\geq$20 years of age, coded as a 1) or children (<20 years of age, coded as a 0). $x_3$ represents a 0 or one value for whether vaccination status is unknown (having a known vaccination status coded as a 0, having an unknown vaccination status coded as a 1), and $x_4$ represents 0 or one value for whether vaccination status is up-to-date (up-to-date coded as a 0 and not up-to-date coded as a 1). Under this formulation, an individual with unknown vaccination status would be coded as $x_3 = 1$, $x_4 = 0$, an individual who is up-to-date would be coded as $x_3 = 0$, $x_4 = 0$, and an individual who is not up-to-date is coded as $x_3 = 0$, $x_4 = 1$. This encoding allows us to evaluate the effects of having an unknown vaccination status and a vaccination status that is not up-to-date.

p-values were assigned via a Wald test, and inferred coefficients were exponentiated to return odds ratios. All codes used to parse the divergence tree and formulate and fit the regression model are available at https://github.com/blab/mumps-wa-phylodynamics/blob/master/divergence-tree-analyses/Regression-analysis-on-descendants-in-divergence-tree.ipynb.

## Rarefaction analysis to estimate transmission clusters

Using the full set of North American mumps sequences, we designated all non-Washington North American sequences as 'background' sequences. We then separated Washington sequences into Marshallese tips (57 total sequences) and non-Marshallese tips (52 total sequences). For this analysis, we excluded the genotype K sequence in our data set due to its extreme divergence from other viruses sampled in Washington, which were all genotype G. For each group (Marshallese vs. non-Marshallese), we then generated subsampled data sets comprised of a random sample of 1 to $n$ sequences, where $n$ is the number of total sequences available for that group. For each number of sequences, we performed 10 independent subsampling trials. Subsampling was performed without replacement. So, for community members, we generated 10 data sets in which one community member sequence was sampled, then 10 data sets in which two community members sequences were sampled, etc. up to 10 data sets in which all 57 community members sequences were sampled. For each subsampled data set, we then combined these subsampled data sets with the background North American sequences, and reran the Nextstrain pipeline. For each subsample and trial, we infer geographic transmission history across the tree and enumerate the number of introductions into Washington. Geographic transmission history was inferred using a discrete trait model in TreeTime (*Sagulenko et al., 2018*). For each number of sequences tested, $n$, we report the number of trials resulting in that number of inferred introductions, and the mean number of inferred introductions across the 10 trials. Each resulting 'cluster' consisted of a set of sequences that are related to one another that descend from a single inferred introduction of mumps into Washington.

## Inference of community transmission dynamics using a structured coalescent model

To infer the rates of migration between community and non-community members and to infer ancestral states of Washington internal nodes, we employed a structured coalescent model. Here, 'state' refers to the inferred ancestral identity of an internal node, where the identity could be inferred as 'Marshallese' or 'not Marshallese'. The multitype tree model (*Vaughan et al., 2014*) in BEAST 2

v2.6.2 (*Bouckaert et al., 2019*) infers the effective population sizes of each deme and the migration rates between them. Because the multitype tree model requires that all partitions contain all demes, we could only analyze four clades that circulated in Washington State and included both Marshallese and non-Marshallese tips. We generated an XML in BEAUti v2.6.2 with four partitions and linked the clock, site, and migration models. We used a strict, fixed clock, set to $4.17 \times 10^{-4}$ substitutions per site year and used an HKY substitution model with four gamma-distributed rate categories. This clock rate was set based on the inferred substitutions per site per year from all North American mumps genomes on nextstrain.org/mumps/na. Migration rates were inferred with the prior specified as a truncated exponential distribution with a mean of 1 and a maximum of 50. Effective population sizes were inferred with the prior specified as a truncated exponential distribution with a mean of 1, a minimum value of 0.001, and a maximum value of 10,000. All other priors were left at default values. In order to improve convergence, we employed three heated chains using the package CoupledMCMC (*Müller and Bouckaert, 2019*), where proposals for chains to swap were performed every 100 states. The analysis was run for 100 million steps, with states sampled every 1 million steps. We ran this analysis three independent times, and combined log and tree file output from those independent runs using LogCombiner, with the first 10% (1000 states) of each run discarded as burn-in. We then summarized these combined output log and tree files. A maximum clade credibility tree was inferred using TreeAnnotator with the mean heights option. To ensure that results were not appreciably altered by the migration rate prior, we also repeated these analyses with migration rates inferred with the prior specified as a truncated exponential distribution with a mean of 10 and a maximum of 50.

Although our complete data set contains approximately equal numbers of sequences from Marshallese and non-Marshallese cases, the four clusters analyzed above are enriched among Marshallese tips. To assess the impact of uneven sampling within these clusters on ancestral state inference, we performed a subsampling analysis. For each cluster, we subsampled down the number of Marshallese tips to be equal to the number of non-Marshallese tips, and reran the analysis as above. While the original analysis used four subclades containing both Marshallese and non-Marshallese tips, one of these subclades only has five tips. Subsampling this particular subtree would have resulted in a subtree with only two tips, thus we excluded this clade from the subsampling analysis. For this sensitivity analysis, the three subsampled data sets had the following tip composition: primary outbreak clade: 26 Marshallese and 26 non-Marshallese tips; 10-tip introduction: three Marshallese and three non-Marshallese tips; 8-tip introduction: four Marshallese and four non-Marshallese tips. We generated three randomly subsampled data sets, and for each one ran three independent chains, with each chain run for 50 million steps, sampling every 500,000. For one of the subsampled data sets, none of the chains converged after 20 days. In each of the remaining two subsampled data sets, two out of three chains converged. We combined these converged chains using LogCombiner, with the first 10% of each run discarded as burn-in. We then summarized these combined output log and tree files, and inferred a maximum clade credibility tree using TreeAnnotator with the mean heights option.

The analysis as described above assumes that each introduction into Washington State is an independent observation of the same structured coalescent process, and that the data set represents a random sample of the underlying population. Additionally, this approach requires a priori definition of which sequences are part of the same Washington State transmission chain. Finally, the above analysis could only make use of the four Washington introductions with both Marshallese and non-Marshallese tips, and excludes other transmission chains. Because of these issues, we supplemented the above approach with an additional analysis using the approximate structured coalescent (*Müller et al., 2017*) in MASCOT (*Müller et al., 2018*). Using all of the Washington sequences, we specified three demes: Marshallese in Washington, non-Marshallese in Washington, and outside of Washington. To account for any transmission that happened outside of Washington State, the 'outside of Washington' deme acted as a 'ghost deme' from which we did not use any samples. The effective population size of this 'outside of Washington' deme then describes the rate at which lineages between any location outside of Washington share a common ancestor. Including specific samples from outside of Washington would bias the inferred effective population size toward the coalescent rates of the sampled locations, by incorporating local transmission dynamics of other locations. We then estimated migration rates and effective population sizes for all three demes, but fixed the migration rates such that the unsampled deme ('outside of Washington') could only act as

a source population. This is motivated by not having observed obvious migration out of Washington State in our previous analysis here. We ran this analysis for 10 million steps, sampling every 5000, and discarded the first 10% of states as burn-in.

## Acknowledgements

We would like to extend our sincerest thanks to Jiji Jally for her help and input on the project. Jiji Jally is an advocate for affordable access to healthcare services, supportive services for the Marshallese community, and works as a translator to assist the community in Washington State. These insightful discussions were absolutely critical for contextualizing our results. We would also like to sincerely thank Kelsey Florek for locating and sharing mumps samples from Wisconsin, Ohio, Missouri, Alabama, and North Carolina, which greatly enhanced the analyses presented here. We also thank Jeff Joy for graciously sharing mumps genomes from British Columbia. Finally, we would like to thank the Fred Hutchinson Cancer Research Center sequencing core for providing excellent sequencing services and Fred Hutch Scientific Computing Infrastructure. LHM is an Open Philanthropy Project fellow of the Life Sciences Research Foundation. AB was supported by the National Science Foundation Graduate Research Fellowship Program under Grant No. DGE-1256082. TB is a Pew Biomedical Scholar and is supported by NIH R35 GM119774-01. Scientific Computing Infrastructure at Fred Hutch is supported by NIH ORIP S10OD028685.

## Additional information

### Funding

| Funder | Grant reference number | Author |
| --- | --- | --- |
| National Science Foundation | DGE-1256082 | Allison Black |
| Life Sciences Research Foundation | | Louise H Moncla |
| National Institutes of Health | R35 GM119774-01 | Trevor Bedford |
| National Institutes of Health | ORIP S10OD028685 | Trevor Bedford |

The funders had no role in study design, data collection and interpretation, or the decision to submit the work for publication.

### Author contributions

Louise H Moncla, Conceptualization, Data curation, Software, Formal analysis, Validation, Investigation, Visualization, Methodology, Writing - original draft, Writing - review and editing; Allison Black, Conceptualization, Data curation, Formal analysis, Validation, Investigation, Visualization, Methodology, Writing - original draft, Writing - review and editing; Chas DeBolt, Conceptualization, Resources, Data curation, Supervision, Writing - review and editing; Misty Lang, Resources, Data curation, Investigation; Nicholas R Graff, Resources, Data curation, Formal analysis, Methodology, Writing - review and editing; Ailyn C Pérez-Osorio, Conceptualization, Data curation, Formal analysis, Methodology; Nicola F Müller, Formal analysis, Visualization, Methodology, Writing - review and editing; Dirk Haselow, Conceptualization, Data curation, Supervision, Project administration, Writing - review and editing; Scott Lindquist, Conceptualization, Supervision, Funding acquisition, Project administration; Trevor Bedford, Conceptualization, Data curation, Software, Formal analysis, Supervision, Funding acquisition, Visualization, Methodology, Project administration, Writing - review and editing

### Author ORCIDs

Louise H Moncla (iD) https://orcid.org/0000-0001-5722-1988
Trevor Bedford (iD) https://orcid.org/0000-0002-4039-5794

### Decision letter and Author response

Decision letter https://doi.org/10.7554/eLife.66448.sa1
Author response https://doi.org/10.7554/eLife.66448.sa2

## Additional files

### Supplementary files

• Supplementary file 1. Sample metadata. All genomes generated for this analysis are described above. Dates are formatted as year-month-day. Vaccination status, Ct, and sample collection type are all available for the Washington samples. Genome coverage represents the total proportion of bases in the genome with at least 20× coverage for which we were able to call a base. Sites with <20× coverage were labeled as Ns. Only samples with at least 50% non-N bases were included in the analysis.

• Transparent reporting form

### Data availability

All code used to analyze data, input files for BEAST, and all code used to generate figures for this manuscript are publicly available at https://github.com/blab/mumps-wa-phylodynamics [https://archive.softwareheritage.org/swh:1:rev:b8358a0d49d70670dbab9eeffa9972c277b3021b]. Raw FASTQ files with human reads removed are available under SRA project number PRJNA641715. All protocols for generating sequence data as well as the consensus genomes are available at https://github.com/blab/mumps-seq [https://archive.softwareheritage.org/swh:1:rev:3309d1535804a71e6d9e7cc55295b6ea61bde730]. Consensus genomes have also been deposited to Genbank under accessions MT859507-MT859672.

The following datasets were generated:

| Author(s) | Year | Dataset title | Dataset URL | Database and Identifier |
|---|---|---|---|---|
| Moncla LH, Black A, DeBolt C, Lang M, Graff NR, Pérez-Osorio AC, Müller NF, Haselow D, Lindquist S, Bedford T | 2020 | Repeated introductions and intensive community transmission fueled a mumps virus outbreak in Washington state - BioProject | https://www.ncbi.nlm.nih.gov/sra/?term=PRJNA641715 | NCBI Sequence Read Archive, PRJNA641715 |
| Moncla LH, Black A, DeBolt C, Lang M, Graff NR, Pérez-Osorio AC, Müller NF, Haselow D, Lindquist S, Bedford T | 2020 | Repeated introductions and intensive community transmission fueled a mumps virus outbreak in Washington state | https://www.ncbi.nlm.nih.gov/nuccore/?term=MT859507%3AMT859672%5Baccn%5D | NCBI GenBank, MT859507-MT859672 |

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

Nomenclature MV. 2020. *Weekly Epidemiological Record Relevé Épidémiologique Hebdomadaire*.

Palafox NA, Riklon S, Alik W, Hixon AL. 2007. Health consequences and health systems response to the Pacific U.S. nuclear weapons testing program. *Pacific Health Dialog* **14**:170–178. PMID: 19772154

Pickett BE, Sadat EL, Zhang Y, Noronha JM, Squires RB, Hunt V, Liu M, Kumar S, Zaremba S, Gu Z, Zhou L, Larson CN, Dietrich J, Klem EB, Scheuermann RH. 2012. ViPR: an open bioinformatics database and analysis resource for virology research. *Nucleic acids research* **40**:593–598. DOI: https://doi.org/10.1093/nar/gkr859, PMID: 22006842

Rambaut A, Lam TT, Max Carvalho L, Pybus OG. 2016. Exploring the temporal structure of heterochronous sequences using TempEst (formerly Path-O-Gen). *Virus Evolution* **2**:vew007. DOI: https://doi.org/10.1093/ve/vew007, PMID: 27774300

Sagulenko P, Puller V, Neher RA. 2018. TreeTime: maximum-likelihood phylodynamic analysis. *Virus Evolution* **4**:vex042. DOI: https://doi.org/10.1093/ve/vex042, PMID: 29340210

Simon SL. 1997. A brief history of people and events related to atomic weapons testing in the Marshall Islands. *Health physics* **73**:5–20. DOI: https://doi.org/10.1097/00004032-199707000-00001, PMID: 9199214

Simon SL, Bouville A, Melo D, Beck HL, Weinstock RM. 2010. Acute and chronic intakes of fallout radionuclides by Marshallese from nuclear weapons testing at Bikini and Enewetak and related internal radiation doses. *Health physics* **99**:157–200. DOI: https://doi.org/10.1097/HP.0b013e3181dc4e51, PMID: 20622550

Snijders BE, van Lier A, van de Kassteele J, Fanoy EB, Ruijs WL, Hulshof F, Blauwhof A, Schipper M, van Binnendijk R, Boot HJ, de Melker HE, Hahné SJ. 2012. Mumps vaccine effectiveness in primary schools and households, the Netherlands, 2008. *Vaccine* **30**:2999–3002. DOI: https://doi.org/10.1016/j.vaccine.2012.02.035, PMID: 22381073

Stack JC, Welch JD, Ferrari MJ, Shapiro BU, Grenfell BT. 2010. Protocols for sampling viral sequences to study epidemic dynamics. *Journal of the Royal Society, Interface* **7**:1119–1127. DOI: https://doi.org/10.1098/rsif.2009.0530, PMID: 20147314

Stapleton PJ, Eshaghi A, Seo CY, Wilson S, Harris T, Deeks SL, Bolotin S, Goneau LW, Gubbay JB, Patel SN. 2019. Evaluating the use of whole genome sequencing for the investigation of a large mumps outbreak in Ontario, Canada. *Scientific reports* **9**:12615. DOI: https://doi.org/10.1038/s41598-019-47740-1, PMID: 31471545

**Takahashi T**, Trott KR, Fujimori K, Simon SL, Ohtomo H, Nakashima N, Takaya K, Kimura N, Satomi S, Schoemaker MJ. 1997. An investigation into the prevalence of thyroid disease on Kwajalein Atoll, Marshall Islands. *Health physics* **73**:199–213. DOI: https://doi.org/10.1097/00004032-199707000-00017, PMID: 9199230

**Towne SD**, Yeary KHK, Narcisse M-R, Long C, Bursac Z, Totaram R, Rodriguez EM, McElfish P. 2021. Inequities in access to medical care among adults diagnosed with diabetes: comparisons between the US population and a sample of US-Residing marshallese islanders. *Journal of Racial and Ethnic Health Disparities* **8**:375–383. DOI: https://doi.org/10.1007/s40615-020-00791-x

**US Census Bureau**. 2021. *Historical Households Tables*.

**Vaughan TG**, Kühnert D, Popinga A, Welch D, Drummond AJ. 2014. Efficient bayesian inference under the structured coalescent. *Bioinformatics* **30**:2272–2279. DOI: https://doi.org/10.1093/bioinformatics/btu201

**Vink MA**, Bootsma MC, Wallinga J. 2014. Serial intervals of respiratory infectious diseases: a systematic review and analysis. *American journal of epidemiology* **180**:865–875. DOI: https://doi.org/10.1093/aje/kwu209, PMID: 25294601

**Volz EM**. 2012. Complex population dynamics and the coalescent under neutrality. *Genetics* **190**:187–201. DOI: https://doi.org/10.1534/genetics.111.134627, PMID: 22042576

**Washington State Legislature**. 2014. *Chapter 246-101 WAC Notifiable Conditions*https://app.leg.wa.gov/wac/default.aspx?cite=246-101

**Williams DP**, Hampton A. 2005. Barriers to health services perceived by Marshallese immigrants. *Journal of immigrant health* **7**:317–326. DOI: https://doi.org/10.1007/s10903-005-5129-8, PMID: 19813297

**Wohl S**, Metsky HC, Schaffner SF, Piantadosi A, Burns M, Lewnard JA, Chak B, Krasilnikova LA, Siddle KJ, Matranga CB, Bankamp B, Hennigan S, Sabina B, Byrne EH, McNall RJ, Shah RR, Qu J, Park DJ, Gharib S, Fitzgerald S, et al. 2020. Combining genomics and epidemiology to track mumps virus transmission in the United States. *PLOS biology* **18**:e3000611. DOI: https://doi.org/10.1371/journal.pbio.3000611, PMID: 32045407

**Wong DC**, Purcell RH, Rosen L. 1979. Prevalence of antibody to hepatitis A and hepatitis B viruses in selected populations of the South Pacific. *American journal of epidemiology* **110**:227–236. DOI: https://doi.org/10.1093/oxfordjournals.aje.a112807, PMID: 224696

**Yamada S**, Dodd A, Soe T, Chen TH, Bauman K. 2004. Diabetes mellitus prevalence in out-patient Marshallese adults on Ebeye Island, Republic of the Marshall Islands. *Hawaii medical journal* **63**:45–51. PMID: 15072347

