## [Decision Letter]

**Acceptance summary:**

This interesting phylogenetic analysis of a mumps outbreak in Washington will be of interest to a wide audience, especially those working at the intersection of pathogen genomics and public health. An array of classic and novel phylogenetic approaches supports the conclusions that mumps was introduced several times in Washington during the outbreak, and that the Washington Marshallese community was particularly at risk of mumps infection and transmission despite high vaccination coverage. Consultation with a community health advocate from the affected communities helps contextualize the results.

**Decision letter after peer review:**

Thank you for submitting your article "Repeated introductions and intensive community transmission fueled a mumps virus outbreak in Washington State" for consideration by *eLife*. Your article has been reviewed by 3 peer reviewers, one of whom is a member of our Board of Reviewing Editors, and the evaluation has been overseen by Miles Davenport as the Senior Editor. The reviewers have opted to remain anonymous.

Essential revisions:

1. Please rephrase conclusions regarding the role of age and vaccination to indicate the effect is inconclusive given the low power of the study/small sample size.

2. Figure 2 is difficult to read. A different color scheme may make this figure more understandable.

3. Please compare the age distribution of sampled cases vs outbreak cases in either supplementary table 2 or 3, this would help better convince readers that the sample is representative of the outbreak cases.

4. Elaborate on the decision to include tips on an internal node as the measured variable as opposed to tips with a branch length of a certain distance.

5. Rephrase heading on line 338

6. Additional contextualization of the problem of sampling bias and how this approach is different may help strengthen the methodology presented in the paper.

7. There were several other mumps outbreaks in the United States in the same 2016-2017 time period. Some discussion about Washington state particularities that prevented mumps transmission outside of the Marshallese community would be warranted.

*Reviewer #1 (Recommendations for the authors):*

I cannot comment much on the sequencing and phylogenetic analyses as this is not my area of expertise, but I can comment on the epidemiological analyses:

• There is an overreliance on p-values in the interpretation of the logistic regression in Table 1. There is no consideration of the fact that the study does not have an adequate sample size to assess the effects of variables other than community status. I would recommend the authors do a post-hoc sample size power calculation, not necessarily to put in the article, but to convince themselves of whether they have the statistical power to assess the impact of vaccination and age. Basically, their logistic regression model is suggesting that in fact we cannot exclude the possibility that vaccination status and age may potentially have very strong effects on transmission (point estimate ORs of 2.21, 0.46, or 0.30, in other words over a doubling or a halving of the odds, which would be considered by many to be an important effect size). The only reason that the analysis has enough statistical power for community status is because the odds ratio is enormous and the prevalence of Marshallese samples is very high. Please read Amrhein V, Greenland S, McShane B. Nature. 2019 Mar;567(7748):305-307 for further discussion on this topic. I would suggest addressing this issue by:

– Mentioning low statistical power to assess the effects of age and vaccination as a limitation of this analysis

– When discussing the results for age and vaccination status, I would recommend using the suggestion of Amrhein et al. and to discuss the point estimate and confidence intervals rather than commenting on their non-statistical significance. (e.g. instead of "Neither age nor vaccination status were significantly associated with the presence of downstream tips in the tree", say instead that while those with unknown vaccination status were more likely to have descendants in the tree than those with known up-to-date vaccination, the confidence interval could not exclude a null or positive association between vaccination status and having tree descendants).

– The conclusion the authors should reach in the abstract and discussion is not that "Neither vaccination status nor age were strong determinants of transmission", but rather that while age and vaccination status might be associated with transmission, the authors did not have enough statistical power to estimate their effect (i.e. the confidence intervals are very wide and cannot exclude either no association or inverse associations).

• I find the parameterization of age very unusual, in general what is more often used for continuous variables is centering the variable around the mean or mode rather than normalizing. I am not convinced that age as a continuous variable is the best way to parameterize age for this analysis. There is no reason to believe that age would necessarily have a linear effect on transmission. Social networks and contacts do not change linearly with age but probably in a more segmented manner as individuals age into different life stages (school, workforce, retirement). Waning vaccine immunity is also possibly not linear with age. This is just a suggestion, but I think it would potentially be more fruitful and maybe yield higher statistical power in this case to categorize age dichotomously as either children (<20y) or adults (>=20y) based on different social networks in children vs adults. However, there may be other cutpoints that fit the data better and that could be explored. Otherwise, some justification for the parameterization of age would be nice.

• Please compare the age distribution of sampled cases vs outbreak cases in either supplementary table 2 or 3, this would help better convince readers that the sample is representative of the outbreak cases.

• Figure 2: Please make the data for Washington state more obvious, for example by restricting an entire color shade such as blue. Currently it is very difficult to distinguish between Georgia, Virginia, Manitoba, Washington, Massachusetts, and North Dakota. It is also not clear from the legend what a grey node represents.

*Reviewer #2 (Recommendations for the authors):*

The input from community health advocates is a strength that could be better highlighted in the discussion. It is unclear which potential causes (possibly all of them) are corroborated by the lived experiences of the individuals in the Marshallese community in WA.

The new test for descendants in the divergence tree is a useful method that is likely to be adopted in other studies. It would be nice if the authors could elaborate very briefly on the decision to include tips on an internal node as the measured variable as opposed to tips with a branch length of a certain distance. Is this decision based on the substitution rate and serial interval of Mumps and if applied by others should it be adjusted based on the pathogen and/or sampling scheme (e.g. Intensely sampling a super spreading event would show a large number of tips of one type at the base of 1 transmission chain which is different than finding many tips with descendants spread throughout the tree)?

The wording in the heading on line 338 is confusing.

Is it known, or suspected from interviews, if members of the Marshallese community are more closely connected through social contacts with each other than they are with the wider public? This seems like an implicit assumption, but very likely given the outbreaks in WA and AR.

Lines 667-668 in the methods mention 27 "states" were used in the phylogeographic reconstruction. The main text and Figure 1 mention 26.

---

## [Author Response]

Essential revisions:1. Please rephrase conclusions regarding the role of age and vaccination to indicate the effect is inconclusive given the low power of the study/small sample size.

We thank the reviewers and editor for this critique, as it was a helpful set of comments. We have rephrased our conclusions throughout the paper. Given a comment from reviewer 2, we have also opted to rename the analysis such that tips are now labelled as either “basal” or “terminal” rather than having descendants or not having descendants. We have altered the following text accordingly:

In the abstract, lines 26-28 have been changed from “Neither vaccination status nor age were strong determinants of transmission” to “Although age and vaccination status may have impacted transmission, our dataset could not quantify their precise effects.”

In lines 250-271, we have rewritten the paragraph to now read:

“When we encounter a tip that lies on an internal node, we enumerate the number of tips that descend from its parent node. […] These results suggest that community membership was a significant determinant of sustained transmission while controlling for vaccination status and age.”

On lines 448-451 of the discussion, we have replaced “We found no support for age or vaccination status as critical determinants of transmission in our outbreak, consistent with epidemiologic findings” to now read “We were unable to evaluate the precise effects of age and vaccination status on transmission in our outbreak. Future, larger studies will be necessary to disentangle the interplay between contact patterns, waning immunity, and vaccination status during mumps transmission.”

We have also added an additional supplemental table (Supplementary File 1d) that lists the number of tips labeled as basal or terminal for each metadata category. This should help clarify exactly how many data points were compared for this analysis, and makes clear that the number of compared data points is indeed small for some categories.

2. Figure 2 is difficult to read. A different color scheme may make this figure more understandable.

We thank the reviewers and editors for pointing this out. We have reorganized Figure 2 and its associated supplements to reduce the number of colors shown. We grouped US states and Canadian provinces together by geography into 6 groups, as suggested by reviewer 2. We grouped US states as follows: non-Washington west USA (California and Montana), south USA (North Carolina, Arkansas, Louisiana, Texas, Virginia, Georgia, and Alabama), Northeast USA (New York, Massachusetts, Pennsylvania, New Hampshire, and New Jersey), midwest USA (North Dakota, Kansas, Missouri, Iowa, Wisconsin, Indiana, Michigan, Ohio, and Illinois), British Columbia Canada, and Manitoba and Ontario Canada. We assigned Washington its own, distinct color so that it can be more easily identified. We now display a total of 7 colors instead of 27, which we hope improves readability. We have also included a new supplemental figure (Figure 2—figure supplement 2) that retains all 27 colors. In this version, each state/province has its own color, with that color representing a shade or tint of the overall region color shown in Figure 2. We included this as a supplement in case there were readers who were interested in the results of the full 27-state geographic reconstruction.

3. Please compare the age distribution of sampled cases vs outbreak cases in either supplementary table 2 or 3, this would help better convince readers that the sample is representative of the outbreak cases.

This is a helpful suggestion, and we have added the counts and percentages for age groups in our dataset to Supplementary File 1b.

4. Elaborate on the decision to include tips on an internal node as the measured variable as opposed to tips with a branch length of a certain distance.

We thank the reviewers and editor for this comment. We have reorganized the portion of the methods that describe this transmission test and divided it into 2 sections: 1. “Quantifying transmission in divergence trees using basal and terminal tips: formulation and rationale” and 2. “Regression model for quantifying a tip’s probability of being basal”. Section one outlines the procedure and includes new paragraphs describing our rationale for defining the cutoff as we did, as well as how we expect this metric to fare for viruses with higher or lower mutation rates. We also include figures to help clarify our assumptions. Section 2 outlines the regression model details. The implementation here was an initial step in developing this method, and fully exploring all interactions among mutation rate, serial interval, sampling intensity and effect sizes is beyond the scope of the current work. We hope these new methods clarify our thought process and provide some intuition for others who may wish to adapt this method to their own work. This new information is written in the Methods on lines 798-877:

“Quantifying transmission in divergence trees using basal and terminal tips: formulation and rationale

To determine whether specific groups were more likely to be part of sustained, serially sampled transmission chains, we developed a statistic to quantify transmission in the tree. […] Future work will be necessary to define the precise interaction between mutation rate, serial interval, sampling intensity and effect sizes in determining the power of this test.”

5. Rephrase heading on line 338

We apologize for this typo, and agree this was confusing. This has been changed to read, “Viruses infecting individuals in different vaccination groups are genetically similar.”

6. Additional contextualization of the problem of sampling bias and how this approach is different may help strengthen the methodology presented in the paper.

We agree that this is a helpful addition, and have added a new paragraph devoted to a discussion of sampling bias to the discussion on lines 458-484. This paragraph reads:

“Sampling bias presents a persistent problem for phylodynamic studies that can complicate inference of source-sink dynamics (De Maio et al., 2015; Dudas et al., 2018; Frost et al., 2015; Kühnert et al., 2011; Lemey et al., 2020; Stack et al., 2010). […] By combining careful sample selection with overlapping approaches to evaluate sampling bias, we were able to mitigate concerns that our source-sink reconstructions are driven by sampling artifacts.”

7. There were several other mumps outbreaks in the United States in the same 2016-2017 time period. Some discussion about Washington state particularities that prevented mumps transmission outside of the Marshallese community would be warranted.

We thank the reviewers and editor for this comment, and agree that further contextualization would be helpful. We did not make it clear in the initial submission that in 2016/2017, the vast majority of mumps outbreaks in the US were associated with either universities or ethnic communities. We have re-organized a few paragraphs in the Discussion section and added information about other 2016/2017 outbreaks. This new paragraph is on lines 499-519, and reads:

“Our finding that most introductions sparked short transmission chains suggests that mumps did not transmit efficiently among the general Washington populace. […] Future work to quantify the interplay between contact rates and vaccine-induced immunity among different age and risk groups should be used to guide updated vaccine recommendations.”

We also amended lines 42-46 in the introduction to highlight that most other US outbreaks in 2016/2017 were university-associated:

“Like with other recent mumps outbreaks, most Washington cases in 2016/17 were vaccinated. Unusually though, while most US outbreaks in 2016/2017 were associated with university settings (Albertson et al., 2016; Bonwitt et al., 2017; Donahue et al., 2017; Golwalkar et al., 2018; Shah et al., 2018; Wohl et al., 2020), incidence in Washington was highest among children aged 10-18 years, younger than expected given waning immunity.”

Reviewer #1 (Recommendations for the authors):I cannot comment much on the sequencing and phylogenetic analyses as this is not my area of expertise, but I can comment on the epidemiological analyses:• There is an overreliance on p-values in the interpretation of the logistic regression in Table 1. There is no consideration of the fact that the study does not have an adequate sample size to assess the effects of variables other than community status. I would recommend the authors do a post-hoc sample size power calculation, not necessarily to put in the article, but to convince themselves of whether they have the statistical power to assess the impact of vaccination and age. Basically, their logistic regression model is suggesting that in fact we cannot exclude the possibility that vaccination status and age may potentially have very strong effects on transmission (point estimate ORs of 2.21, 0.46, or 0.30, in other words over a doubling or a halving of the odds, which would be considered by many to be an important effect size). The only reason that the analysis has enough statistical power for community status is because the odds ratio is enormous and the prevalence of Marshallese samples is very high. Please read Amrhein V, Greenland S, McShane B. Nature. 2019 Mar;567(7748):305-307 for further discussion on this topic. I would suggest addressing this issue by:– Mentioning low statistical power to assess the effects of age and vaccination as a limitation of this analysis– When discussing the results for age and vaccination status, I would recommend using the suggestion of Amrhein et al. and to discuss the point estimate and confidence intervals rather than commenting on their non-statistical significance. (e.g. instead of "Neither age nor vaccination status were significantly associated with the presence of downstream tips in the tree", say instead that while those with unknown vaccination status were more likely to have descendants in the tree than those with known up-to-date vaccination, the confidence interval could not exclude a null or positive association between vaccination status and having tree descendants).– The conclusion the authors should reach in the abstract and discussion is not that "Neither vaccination status nor age were strong determinants of transmission", but rather that while age and vaccination status might be associated with transmission, the authors did not have enough statistical power to estimate their effect (i.e. the confidence intervals are very wide and cannot exclude either no association or inverse associations).

We thank the reviewers and editor for this critique, as it was a helpful set of comments. Please see point 1 under “Essential revisions” for our response to this set of critiques.

• I find the parameterization of age very unusual, in general what is more often used for continuous variables is centering the variable around the mean or mode rather than normalizing. I am not convinced that age as a continuous variable is the best way to parameterize age for this analysis. There is no reason to believe that age would necessarily have a linear effect on transmission. Social networks and contacts do not change linearly with age but probably in a more segmented manner as individuals age into different life stages (school, workforce, retirement). Waning vaccine immunity is also possibly not linear with age. This is just a suggestion, but I think it would potentially be more fruitful and maybe yield higher statistical power in this case to categorize age dichotomously as either children (<20y) or adults (>=20y) based on different social networks in children vs adults. However, there may be other cutpoints that fit the data better and that could be explored. Otherwise, some justification for the parameterization of age would be nice.

We thank the reviewer for this comment, and agree that binning age makes sense. We re-coded age as a binary predictor, classifying individuals as <20 years old or ≥20 years old. As suggested, this did improve statistical power: out analysis now utilizes 59 tips that are <20 years of age, and 50 tips ≥20 years of age. We coded ≥20 years of age as a 1 in the regression model. The new results are shown in a revised version of Table 1. This re-coding resulted in a narrower estimated confidence interval, although it is still not sufficiently resolved to estimate the impact of age precisely. We have updated the text in the Methods and Results to reflect this re-coding and updated results.

• Please compare the age distribution of sampled cases vs outbreak cases in either supplementary table 2 or 3, this would help better convince readers that the sample is representative of the outbreak cases.

Please see point 3 under “Essential revisions” for our response to this set of critiques.

• Figure 2: Please make the data for Washington state more obvious, for example by restricting an entire color shade such as blue. Currently it is very difficult to distinguish between Georgia, Virginia, Manitoba, Washington, Massachusetts, and North Dakota. It is also not clear from the legend what a grey node represents.

Please see point 2 under “Essential revisions” for our response to this set of critiques.

Reviewer #2 (Recommendations for the authors):The input from community health advocates is a strength that could be better highlighted in the discussion. It is unclear which potential causes (possibly all of them) are corroborated by the lived experiences of the individuals in the Marshallese community in WA.

Our apologies for failing to make this clear the first time, and the reviewer is correct in assuming that it is all of these causes that were brought up in our interviews. To clarify this, we have added lines 530-533 of the discussion to introduce the paragraph about the factors likely contributing to high transmission efficiency within the Marshallese community. This now reads:

“The following paragraph outlines contributing factors that were brought to light during our interviews with a collaborating community activist, along with corroborating citations from the literature. Each of these factors were specifically cited as important and directly stem from our interviews with her.”

The new test for descendants in the divergence tree is a useful method that is likely to be adopted in other studies. It would be nice if the authors could elaborate very briefly on the decision to include tips on an internal node as the measured variable as opposed to tips with a branch length of a certain distance. Is this decision based on the substitution rate and serial interval of Mumps and if applied by others should it be adjusted based on the pathogen and/or sampling scheme (e.g. Intensely sampling a super spreading event would show a large number of tips of one type at the base of 1 transmission chain which is different than finding many tips with descendants spread throughout the tree)?

We thank the reviewer for this comment, and think that providing extra contextualization has improved the manuscript. Please see point 4 under “Essential revisions” for our full response to this set of critiques.

The wording in the heading on line 338 is confusing.

We apologize for this typo, and agree this was confusing. This has been changed to read, “Viruses infecting individuals in different vaccination groups are genetically similar,” which we hope is clearer.

Is it known, or suspected from interviews, if members of the Marshallese community are more closely connected through social contacts with each other than they are with the wider public? This seems like an implicit assumption, but very likely given the outbreaks in WA and AR.

The reviewer is correct in this assumption. The Marshallese are frequently described in the literature as “close-knit” with a strong sense of community, and this notion was confirmed by our collaborating community contact as generally true. It should be noted that the advocate did not describe the community as insular; rather, that individuals within the community interact frequently and share a broader sense of family than the single-family unit typical of broader American culture. We have added the following lines to lines 540-544 of the discussion, which we hope clarify these points:

“The Marshallese community is often described as close-knit, with frequent and close interactions among individuals, a strong sense of community, and a broader sense of family than the single-family unit typical of broader American culture (Barker, 2012; Embassy of the Republic of the Marshall Islands to the United States of America, n.d.). Contacts within the community could therefore be more frequent or intense, which may facilitate transmission.”

Lines 667-668 in the methods mention 27 "states" were used in the phylogeographic reconstruction. The main text and Figure 1 mention 26.

We thank the reviewer for catching this typo. The correct number of states is 27, which has been fixed in the main text.